
**Saline groundwater evolution in Luanhe River Delta, China since**
**Holocene: hydrochemical, isotopic and sedimentary evidence**
Xianzhang Dang[1, 2, 3], Maosheng Gao[2, 4,], Zhang Wen[1,], Guohua Hou[2, 4],
Daniel Ayejoto[1], Qiming Sun[1, 2, 3]
[1]School of Environmental Studies, China University of Geosciences, 388 Lumo Rd, Wuhan, 430074,
China
[2]Qingdao Institute of Marine Geology, CGS, Qingdao, 266071, China
[3]Chinese Academy of Geological Sciences, Beijing, 100037, China
[4]Laboratory for Marine Geology, Pilot National Laboratory for Marine Science and Technology,
Qingdao, 266071, China
*Correspondence to*: Maosheng Gao (gaomsh66@sohu.com), Zhang Wen (wenz@cug.edu.cn)



**Abstract**
Since the Quaternary Period, palaeo-seawater intrusions have been suggested to
explain the observed saline groundwater that extends far inland in coastal zones. The
Luanhe River Delta (northwest coast of Bohai Sea, China) is characterized by the
distribution of saline, brine, brackish and fresh groundwater, from coastline to inland,
with a wide range of total dissolved solids (TDS) between 0.38–125.9 g $L^{-1}$.
Meanwhile, previous studies have revealed that this area was significantly affected by
Holocene marine transgression. In this study, we used hydrochemical, isotopic, and
sedimentological methods to investigate groundwater salinization processes in the
Luanhe River Delta and its links to the palaeo-environmental settings. The isotopic
results ($^2$H, $^{18}$O, $^{14}$C) show that deep confined groundwater was recharged during the
Late Pleistocene cold period, shallow saline and brine groundwater was recharged
during the warm Holocene period, and shallow brackish and fresh groundwater was
mainly recharged by surface water. The results of hydro-geochemical modeling
(PHREEQC) suggest that the salty sources of salinization are seawater and
concentrated saline water (formed after evaporation of seawater). The $^{18}$O-Cl
relationship diagram shows that saline and brine groundwater are formed by three
end-member mixings (seawater, concentrated saline water and, fresh groundwater).
In contrast, brackish groundwater is formed after the wash-out of saline groundwater
by surface water. Using palaeo-environmental data from sediments, we found that
palaeo-seawater intrusion during the Holocene marine transgression was the primary
cause of groundwater salinization in the study region. Seawater was found to



evaporate in the lagoon area during the progradation of the Luanhe River Delta; the
resulting concentrated saline water infiltrated into the aquifer, eventually forming
brine groundwater due to salinity accumulation. Surface water recharge and irrigation,
on the other side, would gradually flush the delta plain's saline groundwater. This
study provides a better understanding of saline groundwater evolution in other similar
coastal zones.



## 1 Introduction

It is estimated that 20–40 % of the world's population lives in coastal areas. (Small and Nicholls, 2003; Martinez et al., 2007; UN Atlas, 2010). Groundwater is the primary source of fresh water in this region (Cary et al., 2015). However, groundwater salinization poses a significant threat to everyday living and development activities (Cost Environment Action 621, 2005; de Montety et al., 2008). In recent decades, groundwater salinization in coastal zones are widely concerned and studied. On the one hand, seawater intrusion due to groundwater pumping is a vital salinization process in the coastal aquifer (Reilly and Goodman, 1985; Werner, 2010; Han and Currell, 2018). On the other hand, groundwater salinization caused by the palaeo-seawater intrusion, in response to the Quaternary changes in global sea-level, has been reported in many coastal zones worldwide (Edmunds, 2001; Akouvi, 2008; Santucci et al., 2016).

Coastal aquifers are linked to the ocean and continental hydrological cycle (Ferguson and Gleeson, 2012), both of which are influenced by natural and human-induced change (Jiao and Post, 2019). There is a steady-state seawater-freshwater interface under the natural state that extends inland from the coastal line (Costall et al., 2020). Overexploitation of groundwater locally decreases the land groundwater head, shifting the interface downstream and causing salinization of the freshwater aquifer, which is a phenomenon influenced by human factors (Werner et al., 2013). Since the Quaternary period, sea-level fluctuations on geological timescales have caused the interface to change, allowing seawater intrusion



during transgression events and freshwater flushing during glacial low sea-level
periods, which are the primary factors influencing groundwater quality in coastal
areas (Kooi et al., 2000; Sanford, 2010; Aquilina et al., 2015; Lee et al., 2016).
However, the hypersaline groundwater found in coastal zones, particularly brine
groundwater with a salinity of 2–4 times that of seawater, cannot be explained solely
by using a seawater intrusion model (Sola et al., 2014, Han et al., 2020), and
palaeoenvironment settings must be taken into consideration (Van Engelen et al.,
2019). Some studies, for example, attribute the presence of brine in Mediterranean
countries to the evaporation of seawater in the lagoon system during the Holocene
transgression (Giambastiani et al., 2013, Vallejos et al., 2018).
Deceleration of sea-level rise since the mid-Holocene has resulted in the formation
of global deltas (Stanley and Warne, 1994). Meanwhile, various salinity
palaeo-saltwater has been found in these delta aquifers at distances up to 100 km from
current coastlines (Larsen et al., 2017). Hydrogeochemical, isotopic methods (Wang
and Jiao, 2012; Geriesh et al., 2015; Tran et al, 2020) and numerical simulations (Tran
et al., 2012; Delsman et al., 2014; van Engelen et al., 2018) were used to illustrate the
origin of the inland saline groundwater. Few studies examine the response of saline
groundwater evolution to the palaeoenvironment development. The Luanhe River
Delta, situated on the northwest coast of Bohai Sea, is an independently developed
Holocene coastal delta, with fresh, brackish, saline, and brine groundwater distributed
from land to sea in the shallow aquifer (Dang et al., 2020). In this study, we used a
range of chemical and isotopic indicators to determine the salinity sources and



recharge condition. Using sedimentary evidence from the reported cores, we have
been able to identify groundwater salinization processes and the genesis of brine
which had been subjected to complex climate, geomorphological and hydrological
evolution. This research can be used to better understand saline groundwater
evolution in other coastal zones and, as a result, better manage groundwater resources.
**2 Background of the study area**
The study area is located in northeastern Hebei Province, China, on the northwest
coast of Bohai sea (Fig. 1a). The study area consists of alluvial fan and coastal delta,
bounded by Holocene maximum transgression line (Xue et al., 2016). The delta area
can be further divided into two parts: old delta between the Douhe River and the Suhe
River, the new delta between the Suhe River and the modern Luanhe River (He et al.,
2020). The geomorphology of the study area is inclined to the south and southwest
with a slope of about 0.04–2 ‰. The temperate monsoon climate affects the average
annual temperature of 12.5 °C and annual rainfall of 601 mm (1956–2010), with 80 %
of the annual rainfall occurring between July and September..
**2.1 Hydrogeology**
The thickness of Quaternary sediments in the study area is about 400–500 m.
According to the lithology and hydrogeological characteristics, the Quaternary
aquifers are made up of four distinct aquifers (Fig. 1b): the First Holocene aquifer ($Q_4$)
is a phreatic or semi-confined aquifer with a bottom depth of 15–30 m and is
primarily composed of fine sand and slit. The second Late Pleistocene aquifer ($Q_3$),
the third Middle Pleistocene aquifer ($Q_2$), and the fourth Early Pleistocene aquifer


($Q_1$), with bottom depths of 120–170 m, 250–350 m, and 350–550 m, respectively.
They have confined aquifers primarily made up of medium sand and gravel (Niu et al.,
2019). The first aquifer is mainly recharged by meteoric precipitation and lateral
infiltration of surface water (Li et al., 2013). In the alluvial fan areas, the groundwater
from the first aquifer is widely extracted for irrigation. The largest salt farm in north
China, the Daqinghe Salt Farm, uses shallow brine groundwater for salt production in
the delta area, where agricultural activities are small. The second, third, and fourth
aquifers are mainly recharged by a surrounding mountain range and mainly
discharged by human pumping (Ma et al., 2014).

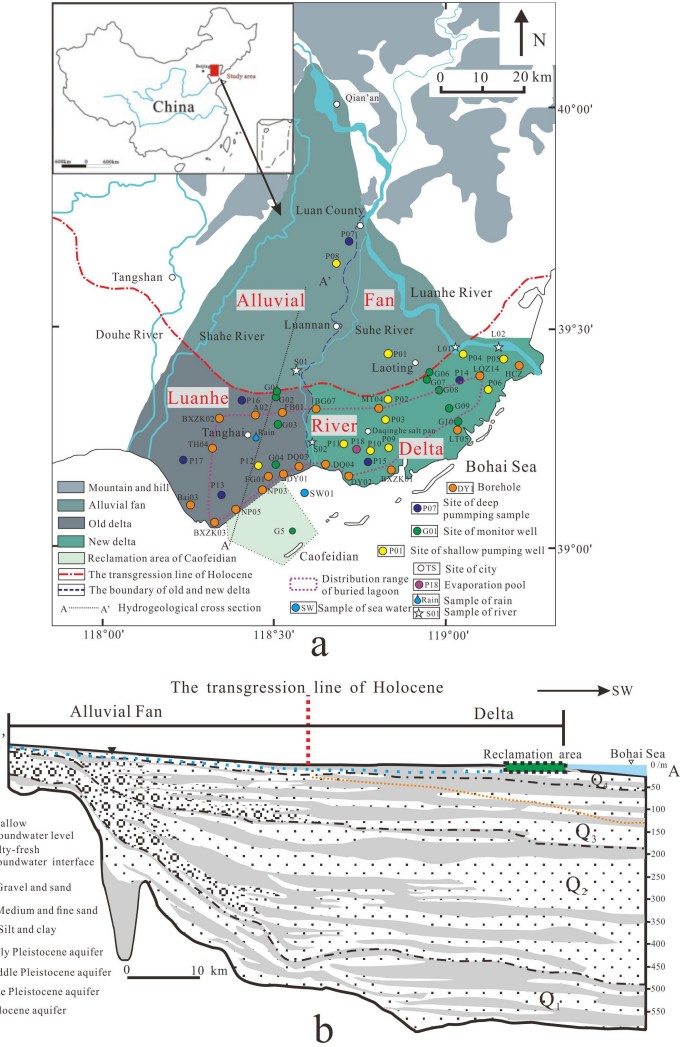

Fig. 1. (a) Location map of study area. Also shown are the sampling site and published cores in the
Luanhe River Delta. Cores LT05, HCZ, BXZK01, BXZK02 and BXZK03 were cited from He et
al. (2020); Cores NP05, NP03, DY01, DQ03, DQ04, DY02, MT04, BG07, FB01, A02 and TH04
were cited from Xu et al. (2020); Core LQZ04 was cited from Cheng et al. (2020); Core FG01 was
cited from Xu et al. (2011); Core Bai03 was cited from Li and Wang. (1983); Core HCZ was cited
from Peng et al. (1981). (b) Hydrogeological cross-section (A-A' in Fig. 1a) of study area,
modified by Ma et al., 2014.





## 2.2 Marine transgression history

During the Quaternary period, there were several times of marine transgression-regression events in the Bohai basin (Wang et al., 1981, Xu et al., 2018), which were significantly influenced by neotectonics (Liu et al., 2016). The published cores show that the Holocene marine or paralic deposits are widely involved in the study area (Peng et al., 1981; Li et al., 1982; Xu et al., 2020). Furthermore, the MIS5 marine deposits are observed in core FG01 at 80–105 m (Xu et al., 2011) and core Bai03 at 33–46 m (Li and Wang, 1983), both of which are close to the shoreline (Fig. 1a). Except for the sediments at a depth of 40 m in core FG01 (Xu et al., 2011), few studies report MIS3 marine deposits in the study area. Moreover, the inland core BXZK02 (Fig. 1a) is clearly lacking MIS3 marine deposit, provided that MIS5 marine deposits are involved at a depth of 23.3–27.2 m, but the upper sediments were fluvial deposit (about 90–30 ka B.P. age) at 13–23 m deep (He et al., 2020). In conclusion, seawater invaded the study area during Holocene marine transgression; MIS3 and MIS5 marine transgression once reached this area, and seawater may have flooded the land area during MIS5 marine transgression, however, MIS3 marine transgression was less dominant and may not have reached the modern land area.

## 2.3 Sedimentary evolution since the Late Pleistocene

Previous studies have shown that in the study region, the interface of salt-fresh groundwater gradually deepens from land to sea, as shown in Fig. 1b, with salt groundwater primarily occurring in the first aquifer of the delta area. According to stratigraphic transect along the present coastline (Fig. 2), the series stratigraphic





architecture of the first aquifer consists of Late Pleistocene continental

facies-Holocene marine facies-Holocene sea-land transition facies (delta

facies)-modern continental facies or artificial fill, indicating that the sediments of the

first aquifer had been deposited from lowstand continental accumulation to marine

transgression and high stand progradation since the Late Pleistocene.

The seawater had not reached the modern coastline from the Last Glacial

Maximum to the early Holocene (about 30–9 ka B.P.). The Luanhe alluvial fan was an

activity in this period (He et al., 2020). Since about 9000 a B.P., the Holocene marine

transgression had approached the present coastline (Xu et al., 2020), and Holocene

marine sediments developed under the sea-level rise condition from 9–7 ka B.P. The

Holocene marine transgression had reached its maximum inland area 20 km from the

modern coastline until about 7 ka BP (Gao et al., 1981; Peng et al., 1981; Xue, 2014,

2016) (Fig. 1 transgression line of Holocene), the accumulation of highstand

prograding delta on top of Holocene marine strata, together with the artificial fill

formed the modern coastal plain. In addition, lagoons are important components of

the Luanhe River Delta (Feng and zhang, 1998). According to the records of lagoon

facies in the published cores in this region, the approximate distribution range of

buried lagoon is shown as a purple dashed line in Fig. 1a.





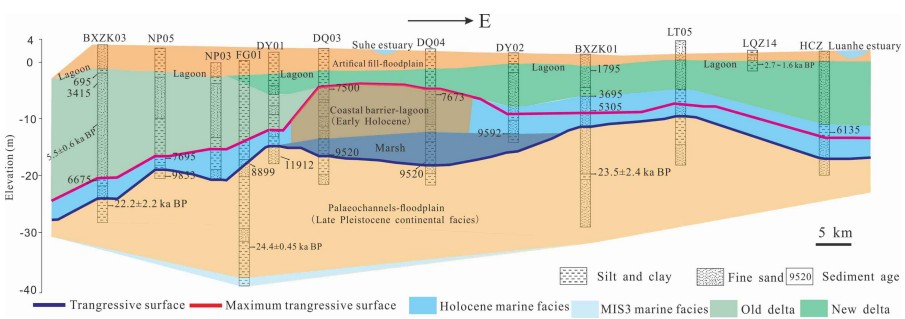

Fig.2 Stratigraphic transect along the present coastline of Luanhe River Delta, modified from He
et al.,2020.
**3 Methods**
In total, 45 water samples were collected from the Luanhe River Delta, including 38
groundwater samples, 5 surface water samples, 1 local rain water and 1 Bohai
seawater samples, during 4 sampling campaigns from October 2016 to June 2020.
Groundwater samples were divided into shallow groundwater samples and deep
groundwater samples, which were pumped from unconfined aquifer and confined
aquifer respectively. The ten monitoring wells are made from 160 mm (internal
diameter) PVC, with slotted screens surrounded by a sand filter pack below bentonite
seals, that tap into shallow aquifer (including upper phreatic aquifer (-20 to 0 m) and
deeper semi-confined aquifer). The screened intervals in monitoring wells are
between -4 to -30 m (G01, G02, G06 and G07), -4 to -50 m (G03, G04, G08 and G09)
and -4 to -80 m (G05 and G10). All wells were fully purged and allowed to recover at
least once prior to survey (sampling and/or EC profiles measuring) which occurred
after stabilization of physio-chemical parameters such as pH, EC. The screened
interval of pumping wells (P01 to P17) which are used for production encompasses





aquifer thicknesses of approximately 5 m above the depths indicated in Table 1.
Surface water includes 2 Suhe River water samples and 2 Luanhe River water
samples. Due to artificial fill that has modified the coastal landscape, it was difficult
to locate the modern lagoon environment. However, during the investigation, it was
found that the Daqingher salt farm in this area extracts seawater into the evaporation
pond. The mixture of seawater and meteoric water is subject to evaporation to form
concentrated saline water (CSW) in the pond, which is similar to the formation of
CSW in a coastal lagoon (Stumpp et al., 2014). Thus, 1CSW in the evaporation pond
was collected.
Water types were classified according to Zhou (2013): freshwater (TDS < 1 g L$^{-1}$),
brackish water (TDS = 1 to 3 g L$^{-1}$), saline water (TDS = 3 to 50 g L$^{-1}$), and brine
(TDS > 50 g L$^{-1}$). Groundwater sampling depths and pH values were measured on site
using CDT-divers. The concentrations of K$^+$, Na$^+$, Ca$^{2+}$, Mg$^{2+}$, and Br$^-$ ion were
measured using inductively coupled plasma analysis (ICAP-7400), while SO$_4^{2-}$ and
Cl$^-$ ions were determined using ion chromatography (ICS-600). The HCO$_3^-$
concentrations of samples were measured using titration. The hydrochemical data are
listed in Table 1. The stable isotope concentrations ($\delta$D, $\delta^{18}$O) of the water samples
(including G02-10, G06-10, G03-05, G04-40, G05-10, G05-46, G07-27, P07-20,
P08-30, P09-30, P10-30, P11-20, P12-40 P14-15, P07-100, P13-200, P14-300,
P15-150, P16-100, P17-200, P18, LH01, LH02, SH01, SH02, SW01, R1) were tested
at the Experimental & Testing Center of Marine Geology, Ministry of Natural
Resource, China, using High Temperature Pyrolysis-Isotope Ratio Mass Spectrometry.





The values of δ$^{18}$O and δD were calculated with respect to the Vienna Standard Mean
Ocean Water (VSMOW), and the uncertainty for δD and δ$^{18}$O are ±1.0 ‰ and ±0.2 ‰,
respectively. The radioisotope (AMS $^{14}$C) of groundwater samples (P14-300, P15-150,
and P16-100) were measured at the Pilot National Laboratory for Marine Science and
Technology. Stable isotopes (δD, δ$^{18}$O, $^{13}$C) and radioisotope of groundwater samples
(G10-10, G03-20, G04-15, G05-30, G06-15, G07-15, G08-15, G08-40, G09-15,
G09-40, G10-10, G10-30) were analyzed at the Beta Analytic TESTING
LABORATORY, where the δ$^{18}$O and δD values were also calculated with respect to
VSMOW, and the uncertainty for δD and δ$^{18}$O are listed in Table 1. The age of
groundwater was calculated using the following Eq. (1)(Clark and Fritz, 1997):
$$t = -8267 \cdot \ln\left( a_t{}^{14}C \, / q - a_0{}^{14}C \right), \tag{1}$$
where t is the residence time of groundwater in a B.P.; $a_t{}^{14}C$ is the measured $^{14}$C
activity in % of modern carbon (pMC); $a_0{}^{14}C$ is the modern $^{14}$C activity of soil derived;
q is a corrective factor, the corrective factor accounts for the dissolution of calcite,
which is assumed to be free of $^{14}$C and, therefore, dilutes the initial $^{14}$C activity of
aqueous DIC in recharged water. The results of $^{13}$C, $^{14}$C and the uncorrected residence
times are listed in Table 2.
**4 Results**
**4.1 Hydrochemistry**
Except for P13-200 (TDS=1.617 g L$^{-1}$, which is brackish water), all the deep
groundwater samples in the study region are freshwater. Deep groundwater
hydrochemical forms shift from Ca-HCO$_3$ to Na-HCO$_3$ as it moves from land to sea





(Fig. 3). Fresh, brackish, saline, and brine water are all forms of shallow groundwater
(Table 1), and the horizontal interface of salt-fresh groundwater corresponds better
with the maximum Holocene transgression line (see Fig. 1a). The Ca-HCO$_3$ type of
shallow fresh groundwater is primarily distributed in the alluvial fan region, and the
brackish, saline and brine groundwater are almost exclusively sampled from the delta
area. The hydrochemical type of brackish water is complex, including Ca-HCO$_3$,
Na-HCO$_3$, and Na-Cl types, while the saline and brine is single Na-Cl type.
For shallow aquifer, vertically, the upper part (depth of 0-15 m) mainly contains
brackish and low TDS saline groundwater, while the lower part (depth of 20–40 m) is
saline and brine groundwater with high TDS. Moreover, for horizontal distribution of
salinity, the groundwater TDS tends to decrease from west to east, such as the TDS of
saline and brine groundwater TDS generally range from 16.57–125.97 g L$^{-1}$ in old
delta (western delta), while 3.26–52.48 g L$^{-1}$ in the new delta (eastern delta).




Table1 Hydrochemical and stable isotopic data from water samples in study area.

| Water | Position | Site | Label | Depth(m) | K⁺ | Na⁺ | Ca²⁺ | Mg²⁺ | Cl⁻ | SO₄²⁻ | HCO₃⁻ | Br⁻ | TDS (g L⁻¹) | pH | δD(‰, VSMOW) | δ¹⁸O(‰, VSMOW) | δ D | δ ¹⁸O |
|---|---|---|---|---|---|---|---|---|---|---|---|---|---|---|---|---|---|---|
| **Shallow groundwater:** | | | | | | | | | | | | | | | | | | |
| | Old delta | G01 | G01-10 | 10.00 | 21.22 | 14.11 | 43.34 | 17.65 | 19.52 | 21.48 | 246.18 | | 0.384 | 7.60 | -58.9 | -8.2 | ±0.39 | ±0.04 |
| | | G02 | G02-10 | 10.00 | 2.39 | 73.33 | 127.27 | 55.26 | 79.74 | 184.82 | 451.77 | | 0.975 | 7.32 | -44.1 | -8.0 | ±1 | ±0.2 |
| Fresh | | P01 | P01-15 | 15.00 | 3.01 | 64.53 | 171.10 | 67.88 | 173.94 | 133.59 | 410.34 | | 1.159 | 8.42 | | | | |
| | Alluvial fan | P07 | P07-20 | 20.00 | 0.973 | 32.2 | 144 | 18.2 | 64.5 | 116 | 199 | 0.074 | 0.575 | 7.78 | -57.1 | -7.5 | ±1 | ±0.2 |
| | | P08 | P08-30 | 30.00 | 1.142 | 13.3 | 53.7 | 16.1 | 14.0 | 23.5 | 151 | 0.037 | 0.384 | 8.20 | -58.8 | -7.9 | ±1 | ±0.2 |
| | | G06 | G06-15 | 10.00 | 5.17 | 209 | 110 | 66.4 | 391 | 204 | 338 | 1.06 | 1.324 | 7.59 | -59.1 | -8.2 | ±0.37 | ±0.08 |
| | | G08 | G08-15 | 10.00 | 35.6 | 666 | 136 | 127 | 1087 | 415 | 374 | 4.38 | 2.841 | 7.84 | -52.6 | -7.3 | ±0.24 | ±0.09 |
| | | G07 | G07-15 | 15.00 | 9.70 | 582.73 | 229.08 | 110.97 | 953.31 | 457.90 | 448.50 | | 2.792 | 7.62 | -54.2 | -6.5 | ±0.14 | ±0.09 |
| Brackish | New delta | P02 | P02-20 | 20.00 | 1.99 | 153.60 | 164.13 | 79.93 | 232.34 | 226.83 | 509.28 | | 1.606 | 8.51 | | | | |
| | | P05 | P05-20 | 20.00 | 56.12 | 310.82 | 207.78 | 112.98 | 397.58 | 394.48 | 686.07 | | 2.584 | 8.39 | | | | |
| | | P14 | P14-15 | 15.00 | 37.60 | 811.00 | 143.00 | 137.00 | 1357.00 | 428.00 | 398.00 | 3.94 | 3.312 | 7.89 | -52.9 | -7.5 | ±1 | ±0.2 |
| | | P06 | P06-20 | 20.00 | 19.03 | 532.42 | 41.36 | 26.15 | 435.29 | 142.11 | 593.62 | | 1.987 | 8.34 | | | | |
| | Old delta | G03 | G03-5 | 5.00 | 162.80 | 4558.4 | 336.08 | 698.13 | 7949.85 | 1388.5 | 1476.45 | | 16.57 | 6.98 | -39.8 | -6.7 | ±1 | ±0.2 |
| | | G04 | G04-15 | 15.00 | 194.47 | 13502. | 559.29 | 1725.67 | 25215.25 | 3565.6 | 1034.50 | | 45.797 | 7.25 | -51.2 | -6.5 | ±0.54 | ±0.07 |
| | | G05 | G05-10 | 10.00 | 200.73 | 5167.9 | 337.58 | 640.08 | 9113.11 | 2208.2 | 432.13 | | 18.099 | 7.45 | -29.8 | -4.9 | ±1 | ±0.2 |
| | | | G05-20 | 20.00 | 185 | 5414 | 291 | 673 | 9223 | 1803 | 646 | 38.7 | 18.235 | 7.41 | | | | |
| | | | G05-46 | 46.00 | 229.51 | 6743.2 | 278.53 | 824.20 | 12432.88 | 1700.9 | 995.21 | | 23.205 | 7.44 | -25.6 | -4.1 | ±1 | ±0.2 |
| | | G07 | G07-27 | 27.00 | 27.11 | 2043.8 | 305.71 | 198.94 | 3650.06 | 570.62 | 350.29 | | 7.147 | 7.32 | -64.6 | -8.7 | ±1 | ±0.2 |
| | | G08 | G08-40 | 40.00 | 66.31 | 7371.3 | 1217.2 | 1028.62 | 15073.12 | 2039.9 | 376.48 | | 27.173 | 6.95 | -46.2 | -5.4 | ±0.26 | ±0.07 |
| Saline | | G09 | G09-15 | 15.00 | 27.07 | 1121.8 | 236.66 | 172.54 | 1885.99 | 406.47 | 576.18 | | 4.426 | 7.2 | -45.3 | -5.7 | ±0.55 | ±0.06 |
| | | | G09-40 | 40.00 | 184.61 | 11882. | 539.23 | 1557.19 | 22669.53 | 2900.7 | 608.91 | | 40.342 | 6.77 | -39.4 | -5.2 | ±0.45 | ±0.08 |
| | New delta | G10 | G10-10 | 10.00 | 294.60 | 9221.2 | 354.32 | 1181.33 | 16220.80 | 2683.1 | 674.39 | | 30.629 | 7.11 | -31.4 | -4.0 | ±0.56 | ±0.05 |
| | | P03 | P03-20 | 20.00 | 16.09 | 285.29 | 432.17 | 202.41 | 682.71 | 1139.6 | 314.65 | | 3.258 | 7.24 | | | | |
| | | P04 | P04-30 | 30.00 | 50.98 | 1103.4 | 133.29 | 135.89 | 1775.19 | 258.68 | 548.21 | | 5.056 | 8.38 | | | | |
| | | P09 | P09-30 | 30.00 | 314 | 12267 | 408 | 1469 | 21909 | 2730 | 543 | 88.8 | 39.64 | 7.19 | -37.7 | -4.9 | ±1 | ±0.2 |
| | | P10 | P10-30 | 30.00 | 159 | 13833 | 744 | 2153 | 26270 | 3725 | 586 | 114 | 47.47 | 6.93 | -27.7 | -3.9 | ±1 | ±0.2 |
| | | P11 | P11-20 | 20.00 | 280 | 15377 | 707 | 2147 | 29689 | 3542 | 404 | 119 | 52.146 | 6.90 | -26.3 | -2.7 | ±1 | ±0.2 |
| | Old delta | G03 | G03-20 | 20.00 | 545.21 | 25182. | 948.30 | 3245.41 | 45835.43 | 4308.9 | 622.01 | | 80.688 | 6.65 | -27.8 | -3.2 | ±0.22 | ±0.14 |
| | | | G03-30 | 30.00 | 489 | 23365 | 776 | 3073 | 42871 | 4383 | 525 | 198 | 75.48 | 6.93 | | | | |
| Brine | | G04 | G04-40 | 40.00 | 159.98 | 23056. | 1253.7 | 3507.54 | 48229.65 | 4450.4 | 667.84 | | 81.325 | 6.7 | -43.1 | -6.4 | ±1 | ±0.2 |
| | | P12 | P12-40 | 40.00 | 836 | 39463 | 759 | 4695 | 70961 | 8518 | 489 | 254 | 125.975 | 6.68 | -36.5 | -4.7 | ±1 | ±0.2 |
| | New delta | G10 | G10-30 | 30.00 | 449.23 | 15416. | 437.65 | 1996.27 | 29889.91 | 3358.7 | 933.01 | 97.4 | 52.482 | 6.98 | -22.5 | -2.1 | ±0.22 | ±0.07 |
| **Deep groundwater:** | | | | | | | | | | | | | | | | | | |
| Brackish | Old delta | P13 | P13-200 | 200.00 | 1.34 | 452 | 40.2 | 31.5 | 417 | 120 | 555 | 2.01 | 1.617 | 7.66 | -70.0 | -9.1 | ±1 | ±0.2 |
| | Alluvial fan | P07 | P07-100 | 100.00 | 1.143 | 19.8 | 45.2 | 12.8 | 12.1 | 13.4 | 163 | 0.036 | 0.267 | 8.29 | -57.1 | -7.6 | ±1 | ±0.2 |
| | New delta | P14 | P14-300 | 300.00 | 0.388 | 104.0 | 17.1 | 3.38 | 29.0 | 20.4 | 205 | 0.10 | 0.379 | 8.14 | -71.3 | -9.8 | ±1 | ±0.2 |
| Fresh | | P15 | P15-150 | 150.00 | 0.540 | 116 | 14.5 | 3.40 | 35.0 | 45.2 | 266 | 0.14 | 0.481 | 7.65 | -70.4 | -9.0 | ±1 | ±0.2 |
| | Old delta | P17 | P17-200 | 200.00 | 0.286 | 144 | 6.29 | 1.61 | 24.5 | 72.0 | 241 | 0.089 | 0.489 | 8.48 | -75.5 | -8.6 | ±1 | ±0.2 |
| | | P16 | P16-100 | 100.00 | 0.355 | 42.6 | 41.8 | 12.5 | 5.29 | 28.4 | 235 | 0.021 | 0.365 | 8.13 | -67.8 | -9.0 | ±1 | ±0.2 |
| **Saurface water:** | | | | | | | | | | | | | | | | | | |
| | New delta | LH01 | LH01 | | 6.16 | 27.25 | 68.93 | 25.67 | 35.41 | 123.53 | 199.70 | | 0.492 | 7.95 | -53.5 | -8.4 | ±1 | ±0.2 |
| Fresh | | LH02 | LH02 | | 6.28 | 27.96 | 68.65 | 26.05 | 36.84 | 124.71 | 199.70 | | 0.495 | 8.04 | -52.5 | -7.3 | ±1 | ±0.2 |
| | Old delta | SH01 | SH01 | | 6.54 | 89.27 | 33.93 | 20.67 | 71.18 | 123.17 | 111.31 | | 0.479 | 8.72 | -54.2 | -7.4 | ±1 | ±0.2 |
| | | SH02 | SH02 | | 12.84 | 108.98 | 126.31 | 52.02 | 98.14 | 298.01 | 396.12 | | 1.095 | 7.55 | -51.0 | -6.9 | ±1 | ±0.2 |
| Seawater | Sea | Bohai | SW01 | | 346.43 | 9025.9 | 338.80 | 1077.72 | 16977.15 | 2578.3 | 140.77 | 58.73 | 30.485 | 7.91 | -8.4 | -3.4 | ±1 | ±0.2 |
| Rain | Rain | Rain | R1 | | | | | | | | | | | | -36.2 | -6.7 | ±1 | ±0.2 |
| Brine | Evaporation | P18 | P18 | | 7530 | 73447 | 184 | 28197 | 174567 | 35794 | 634 | 558 | 320.353 | 6.68 | -19.3 | -0.6 | ±1 | ±0.2 |



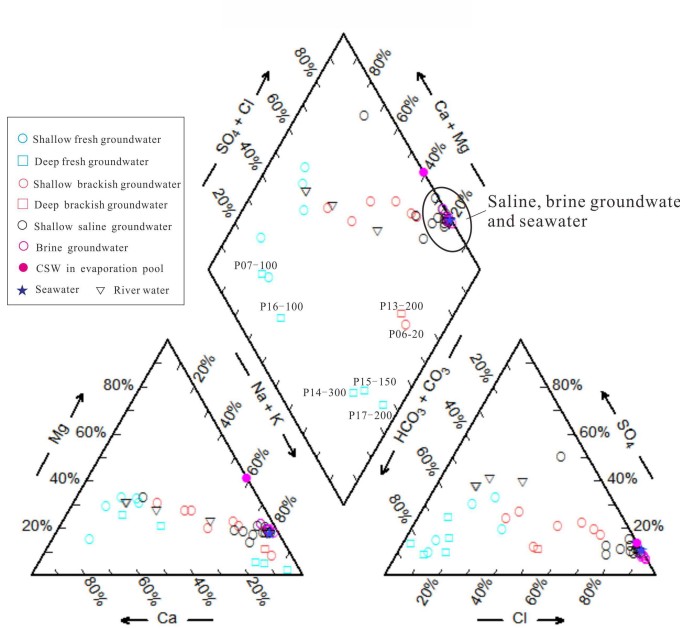

Fig. 3 Piper diagram of the various water samples.
**4.2 $^2$H, $^{18}$O stable isotopes**
Fig. 4 shows the relationship between deuterium and oxygen-18. The global meteoric
water line (GMWL, $\delta^2H = 8 \cdot \delta^{18}O + 10$) is cited from Craig (1961), while the local
meteoric water line (LMWL, $\delta^2H = 6.6 \cdot \delta^{18}O + 0.3$) is based on $\delta^2H$ and $\delta^{18}O$ isotope
data (1985–2003, mean monthly rainfall values) from the Tianjin station, about 100
km southwest of the study area (IAEA/WMO, 2006). The deep groundwater samples
mainly plot in the bottom left of the relationship diagram (Fig. 4), which exhibit
depleted values of stable isotopes, with values of $\delta^2H$ ranging from -75.52 ‰ to
-57.06 ‰ and $\delta^{18}O$ from -9.82 ‰ to -7.61 ‰.
Shallow groundwater samples have higher hydrogen and oxygen isotope levels,
ranging from -64.6 to -22.46 % for $\delta^2H$ and -8.74 to -2.07 % for $\delta^{18}O$. While the
relatively small overall value of fresh and brackish groundwater samples are similar to
those of the river samples, saline and brine groundwater samples showed a much
wider variety (Fig. 4). The shallow groundwater samples, especially brine
groundwater, were generally plotted below the LMWL or GMWL, which mean that
the water was subjected to evaporation prior to recharge into groundwater(Gibson et
al., 1993), or that multiple end-members mixing processes were involved (Han et al.,

7   2011).

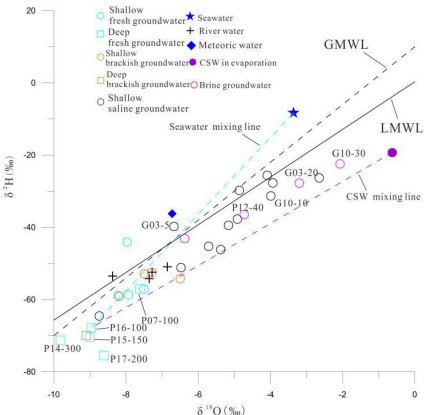

Fig. 4 Stable isotope compositions of different water samples. Seawater mixing line: mixing
between deep fresh groundwater and seawater; CSW mixing line: mixing between deep fresh
groundwater and CSW.
**4.3 Groundwater residence times**
The measured $^{14}$C activities of groundwater samples range from 0.774 to 105.9 pMC
(Table 2). The relationship of $^{14}$C activities and sampling depth is shown in Fig. 5,
which exhibits the overall negative correlations. To estimate the corrective factor q,
two models were used to account for the dissolution of $^{14}$C-free carbon from dissolved
inorganic carbon (DIC) in the aquifer. Since the deep groundwater samples (P16-100,





P15-150, and P14-300) were pumped from a confined aquifer, which is a relatively
closed system, the corrected radiocarbon age was determined using the chemical mass
balance model (CMB) (Clark and Fritz, 1997). For CMB, the q calculated following
Eq. (2):
$q = mDIC_{rech} / mDIC_{final}$,                                                   (2)
where $mDIC_{rech}$ is the DIC molar concentration in the recharging water and $mDIC_{final}$
is the DIC molar concentration in the final groundwater. $mDIC_{final}$ was calculated
following Eq. (3) (Fontes and Garnier, 1979):
$mDIC_{final} = mDIC_{rech} + [mCa + Mg - SO_4 + 0.5(Na + K - Cl)]$,               (3)
$mDIC_{rech}$ was estimated based on groundwater pH and temperature in the assumed
recharge area, e.g., $mDIC_{rech}$ = 10 mmol L$^{-1}$ for pH = 6 and T = 15 °C (Han et al.,

12  2011).

Shallow groundwater samples were collected from semi-confined or phreatic
aquifers, which are semi-open/open system, and thus $\delta^{13}C$ mixing model (Currell et
al., 2010) was used with following Eq. (4) (Pearson and Hanshaw, 1970):
$q = (\delta^{13}C_{DIC} - \delta^{13}C_{CARB}) / (\delta^{13}C_{RECH} - \delta^{13}C_{CARB})$,               (4)
where $\delta^{13}C_{DIC}$ is the measured $\delta^{13}C$ of DIC in groundwater; $\delta^{13}C_{CARB}$ is the $\delta^{13}C$ of
DIC from dissolved soil mineral, using $\delta^{13}C_{CARB}$ = 1.5 ‰ (Chen et al., 2003);
$\delta^{13}C_{RECH}$ is the $\delta^{13}C$ in water when it reaches the saturation zone. Given that the
greater component of C$_4$ vegetation (such as corn) in the study area, $\delta^{13}C_{RECH}$ of
-15 ‰ was used for producing a more realistic set of q values (Currell et al., 2010).
The corrected radiocarbon ages are shown in Table 2, and the residence time of



deep groundwater ranged from 15.9–37.4 cal ka, which is significantly longer than
that of groundwater in the shallow aquifer (about 6.8 cal ka to modern). Moreover, the
ages of brackish and fresh groundwater are modern, while brine has a longer
residence period (about 1.2–4.3 cal ka) and a broader variety of saline groundwater
samples.

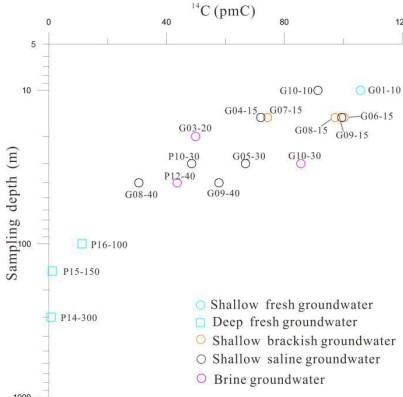

7                     Fig. 5 $^{14}$C activity with sampling depth in groundwater.

Table 2 $^{14}$C measured value and corrected ages of groundwater samples in the Luanhe River Delta.

| Site | Label | $^{14}$C(pmC) | Uncorrected Radiocarbon Age(a B.P.) | δ $^{13}$C (‰, VPDB) | Corrected model | q | Corrected Age(cal a B.P.) |
|------|-------|---------------|-------------------------------------|----------------------|-----------------|---|---------------------------|
| G01 | G01-10 | 105.9±0.40 | Modern | -12.6 | | 0.85 | Modern |
| G03 | G03-20 | 49.9±0.2 | 5590 | -12.6 | | 0.85 | 4323 |
| G04 | G04-15 | 72±0.3 | 2640 | -14.7 | | 0.982 | 2495 |
| G05 | G05-30 | 66.8±0.2 | 3240 | -11.8 | | 0.81 | 1512 |
| G06 | G06-15 | 100.2±0.4 | Modern | -8.2 | | 0.59 | Modern |
| G07 | G07-15 | 74.3±0.3 | 2390 | -10.2 | δ$^{13}$C model | 0.71 | Modern |
| G08 | G08-15 | 97.30±0.4 | 220 | -9.2 | | 0.65 | Modern |
| G08 | G08-40 | 30.6±0.1 | 9510 | -10.4 | | 0.72 | 6884 |
| G09 | G09-15 | 99.5±0.4 | 40 | -11.5 | | 0.79 | Modern |
| G09 | G09-40 | 57.8±0.2 | 4410 | -11.3 | | 0.78 | 2367 |
| G10 | G10-10 | 91.4±0.3 | 720 | -14.3 | | 0.96 | 376 |
| G10 | G10-30 | 85.6±0.3 | 1250 | -15 | | 1 | 1245 |
| P14 | P14-300 | 0.774±0.08 | 39050 | | | 0.83 | 37486 |
| P15 | P15-150 | 1.21±0.09 | 35460 | | CMB model | 0.83 | 33951 |
| P16 | P16-100 | 11.33±0.1 | 17490 | | | 0.82 | 15959 |





## 5 Discussion

### 5.1 Isotopic analysis for origin and recharge of groundwater

Deuterium and oxygen-18 are good tracers for groundwater origin and climatic conditions during recharge periods (Clark and Fritz, 1997). When combined with groundwater residence time, they could further identify modern and palaeo recharge (Han et al., 2014).

The depletion of $^{18}O$ and $^{2}H$ values in the deep fresh groundwater (Fig. 4) can be attributed to a cold climate (Kreuzer et al., 2009) and corrected radiocarbon ages of P15-150 and P14-300 samples (33951 and 37486 cal a B.P., respectively) which may suggest that there was a recharge during the last glacial maximum. Although P16-100 (15959 cal a B.P.) has a slightly higher stable isotope content than deeper groundwater, which is typical of the recharge source as the atmosphere has changed since the last deglaciation (Hendry and Wassenaar, 2000). The stable isotope values of river samples are similar to those of the shallow brackish and fresh groundwater compositions of the modern age, indicating lateral recharge of surface water locally. Meanwhile, in Fig. 4, G03-5 is close to the rainfall sample, indicating that modern precipitation is a new recharge source. The relatively enriched stable isotopic values and radiocarbon data (6883 cal a B.P. to modern) suggest that the brine and saline groundwater formed during warm Holocene, and they were later recharged by surface water (e.g., G09-15 sample with modern age is closed to the river samples in Fig. 4). Additionally, due to mixing of meteoric water, and the subsequent non-equilibrium fractionation of hydrogen isotope during evaporation (Clark and Fritz, 1997), the





CSW sample is characterized by $^{18}$O enrichment compared to seawater but $^{2}$H
depletion.
The marine and lagoon facies indicate that the study region was inundated by
seawater or lagoon water during Holocene sea-level rise, as shown in Fig. 2. The
seawater or lagoon water with enriched stable isotopes would recharge the
isotopically depleted fresh groundwater, explaining that the $\delta\ ^{18}$O value of some saline
and brine groundwater samples trend toward seawater or CSW in Fig. 4. The
significantly high TDS saline groundwater samples, and some brine samples with
reduced stable isotope, fall between the seawater and CWS mixing lines, posing
several end-member mixing processes as to groundwater salinization, which are
further discussed in section 5.3.
**5.2 Hydrochemical analysis for sources of salinity**
As previously stated, during Holocene transgression, seawater will infiltrate into
aquifers(Santucci et al., 2016), causing groundwater salinity to be significantly
affected in the study region . Fig. 6 shows the ion concentrations of various water
samples plotted on a Schoeller diagram. The properties of most saline groundwater
and brine samples are clearly similar to those of seawater, though some samples have
higher concentrations of than that of seawater, implying that the salinity in these
groundwater samples is most likely derived from a marine source.

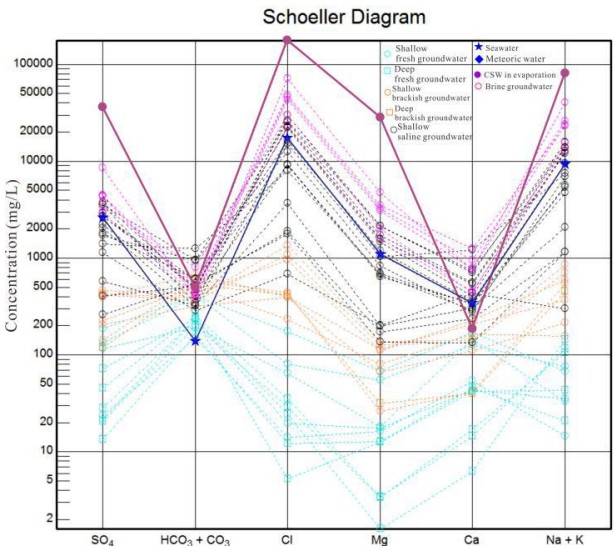

Fig. 6 Schoeller diagram of various water samples.

The PHREEQC code (Parkurst and Appelo, 2013) was used to measure and plot the

theoretical seawater-freshwater mixing line ("mixing line") and seawater evaporation

line ("evaporation line") using hydrogeochemical modeling. Using both simulation

effects as references to groundwater hydrochemical characteristics (Figs. 7 and 8),

which could help to distinguish the sources of groundwater salinity. For the Na-Cl

(Fig. 7a), Mg-Cl (Fig. 7b), and Br-Cl (Fig. 8a) diagrams, whose measured brackish,

saline and brine groundwater samples fit quite well to modeling mixing lines and

evaporation lines follow linear trends from the least to the most saline. This would

strongly demonstrate that, the salinity of salinization groundwater mainly originates

from seawater or, the CSW which is subject to evaporated seawater. In contrast, the

samples deviate from the modeling lines (Fig. 7c and 7d), indicating that there may be

other hydrogeochemical processes responsible for the modified ionic compositions: (1)

Due to reach saturation, there were loss of ions follow mineral precipitation such as





calcite (CaCO₃), gypsum (CaSO₄), and halite (NaCl) during CWS formation, which
consequently explains the decline of $Ca^{2+}$ in P18 and P12 samples in Fig. 7d and,
uplift of Br/Cl ratios in brine samples in Fig. 8b. (2) Calcite and gypsum will be
dissolved along with surface water during lateral recharge, resulting in brackish
groundwater samples plotted above the mixing line, highlighting surface water
flushing processes in the study region. (3) Decomposition of organic matters which
are abundant in marine or lagoon facies sediments can result in release of bromide
ions, and thus making the Br/Cl ratios of saline groundwater samples higher than the
mixing line (Fig. 8b).

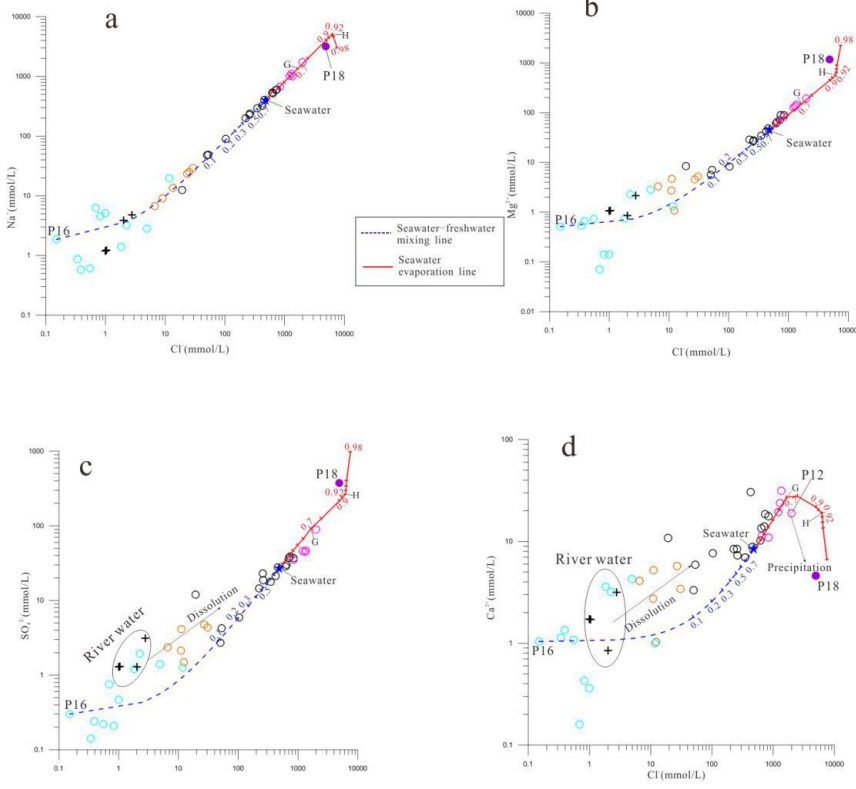

Fig. 7 Hydrochemical relationship between Cl and major ions of measured samples and simulated



results (seawater-freshwater mixing line: theoretical mixing between seawater and deep fresh
groundwater, and the blue numbers are mixing ratios of seawater; seawater evaporation line:
theoretical evaporation of Bohai seawater, and the red numbers are different evaporation rates) in
groundwater. G and H stand for point of precipitation of gypsum, halite respectively. The symbols
of samples are same as Fig. 6.

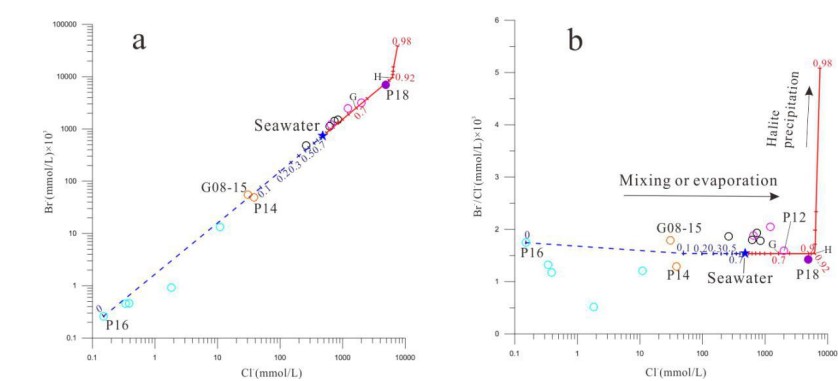

Fig. 8 Relationship between chloride and bromide content in water samples. Symbols are same as
Fig. 7.
**5.3 Mixing processes**
The Concentration of Cl⁻ and $\delta^{18}O$ were widely used to examine the mixing processes
among different end-members in groundwater (Douglas et al., 2000; de Montety et al.,
2008; Liu et al., 2017; Han and Currell, 2018). Fig. 9 depicts the relationship between
$\delta^{18}O$ and Cl⁻ in different water samples. In brine samples, there is a higher Cl⁻
concentration and lower $\delta^{18}O$ values than in seawater, meaning that simple two
end-members mixing cannot adequately explain groundwater salinization. As a result,
the SW01 and P18 were chosen to represent saline end-members, while the P16-100
was chosen to represent fresh end-members that could have been impacted by
infiltration of overlying seawater or CSW during sea-level rise, based on the


hypothesis of three end-member mixing processes. In Fig. 9, an inferred salinization
zone was established that included almost all saline and brine groundwater samples,
demonstrating the salinization processes in which fresh groundwater mixed with
either seawater, CSW, or a mixture of both. The fresh and brackish groundwater
samples, on the other hand, have low Cl$^-$ concentrations and depleted $^{18}$O, deviating
from the assumed salinization zone but approaching the river samples in Fig. 9,
implying a river water-groundwater mixing trend. This trend will wash out the
above-mentioned salinization, owing to lateral recharge of surface water towards the
continental area, which led to a decrease in salinity in groundwater over time, as
shown in the G09-15 sample. In addition, a presumed freshening zone could form
between two river water-groundwater mixing lines, indicating freshening processes in
the Luanhe River Delta that may have been retained since the delta progradation.

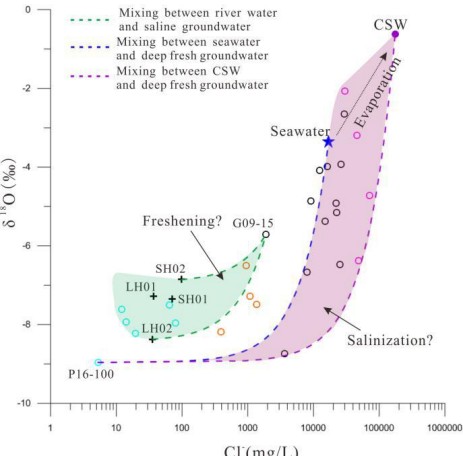

Fig. 9 Relationship between Cl and δ$^{18}$O of different water samples as means to various mixing
processes in the Luanhe River Delta. The symbols are same as Fig. 6. The green area is assumed
freshening zone, and the purple area is assumed salinization zone.



## 6 Interpretation of palaeo-environmental development

Previously, we introduced that the continental area of the Luanhe River Delta is mainly affected by MIS5 and Holocene marine transgression (see 2.3). Assuming that the MIS5 marine transgression event resulted in palaeo-seawater intrusion in the study region. Overlying MIS 5 marine deposits, the evidence of channel deposits in core BXZK02 (dated between 100 ka B.P. and 10 ka B.P., He et al., 2020) and lacustrine deposits in core FG01 (Xu et al, 2011) both imply that fresh surface water flushing the upper saline aquifer would have taken a long time after MIS 5 marine transgression. In addition, since the last deglaciation (about 15 ka B.P.), the palaeo-coast zone has approximately 100 m depth below present sea level along the shelf edge (Li et al., 2014). Stronger river down-cutting and flushing in the study region would have been helped by a large hydraulic gradient and a large shift in palaeoclimate (Xu et al., 2011), resulting in the fresh groundwater found near the core BXZK02 as P16-100 with a corrected radiocarbon age of 15959 cal a B.P., which is likely to provide evidence that the salinization groundwater related to MIS 5 marine transgression could have been flushed out until the Latest Pleistocene. Accordingly, we believe that the observed saline groundwater in the Luanhe River Delta is probably related to the subsequent Holocene marine transgression. This research develops the evolutionary pattern of saline groundwater, as shown in Table 3 and Fig. 10, based on hydrochemical and isotopic analysis, together with the sedimentary evolution of the study region after the Holocene. Three phases are synthesized and reconstructed, as follows.





Table 3 Saline groundwater evolution processes in study area

| Evolution stage | Groundwater evolution processes | | Influencing factors | | | Major hydrogeochemical processes | Sediments |
|---|---|---|---|---|---|---|---|
| | Evolutionary pattern | Factors | Palaeoclimate | Geological setting | Others | | |
| Phase 3 The development of new delta (3. 5ka B. P. to present) | Freshening | Wash-out of surface water | temperate, slightly semi-humid | Development of surface straem | irrigation return | Mixing and leaching | Holocene alluvial deposit or artificial fill Bottom sediments age about 1795—302 a B. P. (Xu et al., 2020 He et al., 2020) |
| | Deceleration of brine formation | Limitations of seawater evaporation | | Diversion of channels and lagoon filled by diluvial deposit | artificial reclamation and offshore levees | | Holocene lagoon facies Bottom sediments age about 5995—1600 a B. P. (Cheng et al., 2020 He et al., 2020) |
| Phase 2 The development of old delta (7 to 3. 5ka B. P. ) | Brine formation | Sseawater evaporation and CSW infiltrating | temperate, slightly arid | Deceleration of sea-level rising, development of delta, and coastal lagoons have been active | Tides or storm | Mixing, leaching, evaporation, and mineral precipitation | Holocene delta facies Bottom sediments age about 6675—3695 a B. P. (He et al., 2020) |
| Phase 1 Holocene transgression (12 to 7ka B. P. ) | Groundwater salinization | Palaeo-seawater intrusion | temperate-warm, humid | Deglaciation of ice sheet, rapid rising of sea level, Holocene transgression | | Mixing | Holocene marine facies Bottom sediments age about 8620—5595 a B. P (Li et al., 1982) Late Pleistoncene continental facies (Xu et al., 2020 He et al., 2020) |

*Phase 1: Transgressive system tract-Holocene transgression stage (9–7 ka B.P.)*
Global sea level was affected by deglaciation of the ice sheet (Fairbanks, 1989),
causing sea level to rise rapidly during the deglaciation period (15.4–7 ka B.P.) (Li et
al., 2014). Previous studies have shown that seawater reached the southwestern Bohai
Bay at 9.9 ka B.P. (Xu et al., 2015), and the present coastline of the study area at
about 9 ka B.P. (Xu et al., 2020), and then the Holocene marine transgression
approached its maximum in the Bohai Sea region at about 7 ka B.P. (Xue 2009, 2014).
It could be summarized that the Holocene transgression stage, which occurred
between 9 and 7 ka B.P, resulted in the study area being inundated by seawater (Fig.
10a). On the one hand, there would have been a tendency for the denser seawater to
infiltrate through the aeration zone and to mix with the fresh groundwater under the
aquifer (Santucci et al., 2016); on the other hand, sea-level rise would cause the
seawater-freshwater interface to move landward (Ferguson and Gleeson, 2012), both
of which contributed to palaeo-seawater intrusion. The characteristics of ionic
components in the salinized groundwater are similar to those of seawater in Fig. 6.
The G08-40 contains TDS of 27.173 g L$^{-1}$, which is more similar to that of SW01.



Simultaneously, the corrected radiocarbon age is 6884 cal a B.P., indicating trapped
palaeo-seawater at low-permeability aquitard sediments still exists and may be
another critical salinity source for neighboring aquifers in the coastal zone (Post and
Kooi, 2003; Lee et al., 2016).

5        The presence of palaeo-seawater intrusion during Quaternary has been recorded in

other coastal regions worldwide (Groen et al., 2000; Bouchaou et al., 2009, Tran et al.,
2012; Han et al., 2020). For the works described above, the salinity of groundwater
after salinization could not exceed that of seawater due to palaeo-seawater intrusion.

9        Other salinization processes that occurred during palaeo-environmental growth are

likely to be correlated with such brine groundwater.
*Phase 2: Highstand system tract-Old Luanhe River Delta development (7–3.5 ka B.P.)*

12        Since about 7 ka B.P. (Saito et al., 1998; Zong, 2004), global marine deltas such as

the Nile Delta, Mississippi Delta, Yangtze Delta, and Luanhe River Delta have
developed (Stanley and Warne, 1994). .

15        Previous research has revealed sediments characteristic of a lagoon environment in

the western study region, indicating that this lagoon was active during the
progradation of the old Luanhe River Delta between 7 and 3.5 ka B.P. (He et al., 2020;
Xu et al., 2020). Meanwhile, after around 5500 B.P., the humid palaeoclimate in this
region has changed to be slightly arid, which may lead to increased evaporation (Jin,
1984). The ancient lagoon would be an ideal location for evaporating seawater that
had been trapped due to storms or tides (Fig. 10b). As a result, concentrated saline
water (CSW) with salinity higher than seawater would have created, and the CSW



kept in the lagoon would go through two processes: (1) infiltrating and descending to
the lower part of the aquifer due to its higher density, and combining with the
salinized groundwater from phase 1, resulting in a three end-members mixing
scenario in the relationship diagram (Fig. 9). (2) After reaching saturation during the
later stages of evaporation, mineral precipitation, such as gypsum, calcite, and halite,
would occur, and this would be subjected to redissolution by meteoric waters or
seawater, resulting in high salinity water that would then be subjected to the above
process; The Br/Cl ratios in certain fresh or brine groundwater samples deviate from
the evaporation line (Fig. 8b), which may be related to halite precipitation and
redissolution. These two processes caused groundwater salinity to rise even further,
resulting in the formation of brine groundwater with 3 times the TDS of seawater,
such as G03-20 with a resident time of 4323 cal a B.P.
*Phase 3: New Luanhe River Delta development (3.5 ka B.P. –present)*
Since about 3500 a B.P., a nearly 90-degree diversion of the Luanhe River channel
in the study area resulted in new delta development (Wang et al., 2007; Xue et al.,
2016). There are some signs of a lagoon environment in the new Luanhe River Delta,
such as core LQZ14 in Fig. 1a, which includes a lagoon deposit with a radiocarbon
date of about 2 ka B.P.(Cheng et al., 2020). And, as previously discussed, the brine
groundwater sample G10-30 would be attributed to evaporation in a lagoon setting
(Fig. 10c). However, some factors are likely to limit the CSW formation in the study
area: (1) the palaeoclimate of the study area changed to semi-humid at about 2.5 ka
B.P. (Jin, 1984), contributing to low evaporation capacity; (2) the diluvial deposit or



artificial reclamation would have filled the coastal low-land such as lagoons, and (3)
offshore levees prevent the seawater from flooding inland during storms or tides.
These factors may also explain why, unlike the old Luanhe River Delta, the current
Luanhe River Delta does not have high TDS brine groundwater.
In addition, under the semi-humid palaeoclimate, some abandoned channels have
developed into small rivers after the diversion of the ancient Luanhe River (Gao,
1981), such as the Suhe River and Shahe River. Firstly, the lateral recharge from the
surface stream plays a role in washing out the salty groundwater. Secondly, due to the
inefficiency of saline groundwater throughout human history, river irrigation has been
commonly used for agricultural activities in the study region, freshening the upper
saline aquifer (Fig. 10c). The brackish and low TDS saline groundwater with modern
age (e.g. G08-15, G09-15), and rapid increase in Electric Conductivity profile (Dang
et al., 2020), are compelling evidence that freshening processes have occurred in the
delta plain, as shown by the $\delta^{18}O\text{-}Cl^-$ relationship diagram (Fig. 9). Some groundwater
samples found above the seawater mixing line in the Ca-Cl and $SO_4$-Cl relationship
diagrams (Fig. 7c, d) may be related to mineral dissolution during river water or
irrigation recharge. However, saline groundwater can be washed out over time in
coastal zones with low-permeable marine layers and a low hydraulic gradient (van
Engelen et al., 2019; Han et al., 2020).
In summary, the evolution of saline groundwater in the study area is a result of
palaeo-environment development such as sea-level change, palaeogeography, and
palaeoclimate, and is significantly affected by human activities. Sea-level rise led to



palaeo-seawater intrusion. After deceleration of sea-level rise, there would be
formation of brine groundwater and slow wash-out during the delta development. The
coastal brine groundwater is a special product of geological evolution, and this study
infers the following conditions for its formation: (1) stable evaporative environments
(e.g. lagoon), (2) suitable climatic conditions (e.g. arid), (3) seawater entering
evaporative environments (e.g. storm or tide), and (4) long-term scale for salinity
accumulation.

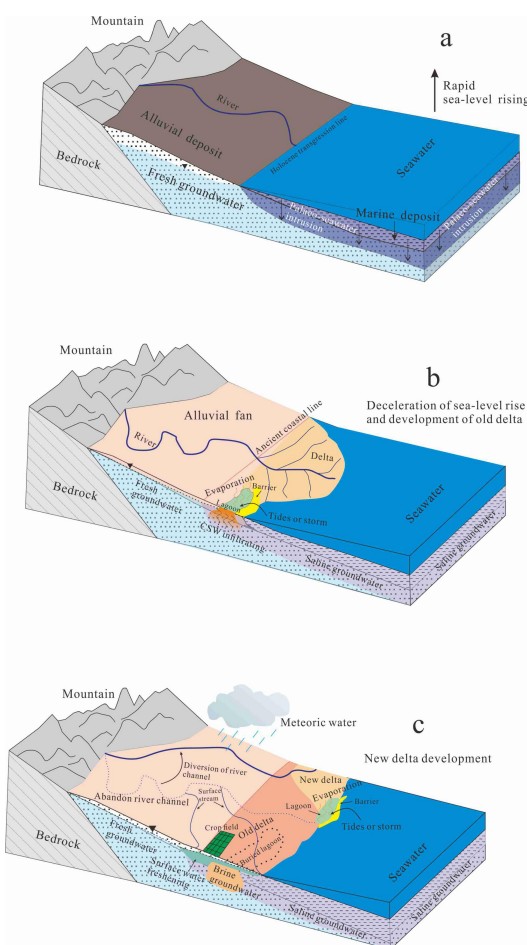

Fig. 10 Diagram of palaeoenvironmental development since Holocene and evolutionary pattern of

10                    saline groundwater.



## 7 Conclusions

The brackish, saline and brine groundwater have been observed at least 20 km inland in the Luanhe River Delta. In this study, we analyze the recharge and salinity source of groundwater, as well as the salinization and freshening processes, using hydrochemical and isotopic methods. The evolution of saline groundwater and its connection to palaeo-environmental settings were studied using sedimentary characteristics as multiple lines of evidence. The following are the key findings:

(1) Different groundwater recharges are identified using environmental isotope analysis ($^2$H, $^{18}$O, $^{14}$C). For the groundwater and Bohai seawater samples, hydrogeochemical modeling (PHREEQC) was used, with a fresh groundwater-seawater mixing line and a Bohai seawater evaporation line as assumptions. The measured and simulated value agrees well, implying that seawater or concentrated saline water is the primary salty source of groundwater salinization. The variation in the $^{18}$O-Cl relationship of multiple water samples further indicates that majority of the saline and brine groundwater originates from three mixing end-members: fresh groundwater, seawater, and concentration saline water. However, there would be some freshening processes observed in brackish groundwater samples, suggesting the wash-out of saline groundwater by surface water.

(2) The evolution of saline groundwater could be reconstructed and summarized using the palaeoenvironmental information contained in the sediments. Given the sea level fell to the lowest position during the Last Glaciation, the palaeochannels downcutting would have contributed to the intense recharge of groundwater by river





water. This study infers that fresh groundwater at upper aquifers before the Holocene marine transgression reached the study area. The evolution of saline groundwater has been traced to three distinct phases: (1) The study area was gradually submerged by seawater around 9–7 ka B.P., and groundwater salinization occurred due to palaeo-seawater intrusion. (2) During the development of the old Luanhe River Delta between 7 and 3.5 ka B.P., the concentration of saline water in the lagoon environment of delta-front continuously provided salinity to the groundwater, and under the effects of evaporation, mixing, and dissolution, some brine groundwater was formed. (3) After the Luanhe River channel's diversion at about 3.5 ka B.P., the new Luanhe River Delta began to develop. On the one hand, the diluvial deposit and human activities limit the formation of brine groundwater; on the other hand, the lateral recharge of surface water and irrigation return would cause partly slow wash-out of saline groundwater in the delta plain.

In coastal zones which similar to this study area, over-extraction of deep groundwater may not only lead the interface of seawater-freshwater to move landward but also cause groundwater salinization by leakage of saline water in adjunct aquifers. If this leak occurs, it will cause widespread salinization of fresh groundwater, particularly if high-salinity brine is presented, which will endanger water quality, like the groundwater salinization in Laizhou Bay. To effectively avoid pollution from saline groundwater movement, this study recommends continuous monitoring of groundwater quality and levels, as well as successful well policies and programs for groundwater resource use.





**Authors contribution**
**Xianzhang Dang:** Conceptualization, Formal analysis, Investigation,
Writing-Original Draft, Data curation.
**Maosheng Gao:** Funding acquisition, Methodology, Supervision, Investigation,
Writing-Review & Editing.
**Zhang Wen:** Supervision, Writing-Review & Editing.
**Guohua Hou:** Project administration, Investigation.
**Daniel Ayejoto:** Writing-Review & Editing.
**Qiming Sun:** Investigation.



**Acknowledgement**
This study was financially supported by the National Natural Science Foundation of
China (41977173), National Key Research and Development Program of China
(No.2016YFC0402800) and the National Geological Survey Project of China
Geology Survey (No. DD20211401). The authors would like to thank Sen Liu,
Chenxin Feng, Chen Sheng, Xueyong Huang and Haihai Zhuang, for their help and
support in collecting field data and conducting geological survey.



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
