# Peer review of "Saline groundwater evolution in Luanhe River Delta, China since Holocene: hydrochemical, isotopic and sedimentary evidence"

_Hydrology and Earth System Sciences, 2021_

## Author Comment (AC1)

**General Comment:**

The authors focus on observed saline groundwater that extends far inland in the coastal zones, and investigate the origin of groundwater salinity and elucidate the major processes controlling salinized groundwater evolution in Luanhe River Delta using hydrochemical and isotopic methods and sedimentary characteristics. The method for interpreting the data is relatively standard. The text is well structured and well supported by the figures. The evolutionary pattern of saline groundwater is innovative, and can be used to better understand groundwater evolution in coastal zone. I would like to recommend the acceptance of the manuscript for publication after minor revisions. Below are my comments that may help with this process.

Response: Thank you for your general comments concerning our manuscript. The manuscript will be further improved and polished carefully for readability and English language. We will implement the following specific comments more in depth.

**Specific comments:**

The introduction provides a comprehensive summary of the groundwater salinization in coastal zones due to Quaternary transgression. The relevant study of similar coastal zones could be expended a little as it is contributed to more valuable scientific significance in International Journals.

Response: We will improve this section to introduce more relevant study in other similar areas globally.

Hydrogeology: this paper is aimed to saline groundwater in coastal aquifer of Luanhe River Delta. However, I cannot find more information about water quality in this section. And how about the hydraulic connection between shallow and deep aquifers? The authors should introduce more details here.

Response: We will add the information of groundwater quality in this section, and further interpret the hydraulic connection between shallow and deep aquifers.

Page 9, line 6, "MIS5". The first abbreviations need to be explained in detail.

Response: The first abbreviations will be re-check, and explained clearly.

Page 9, lines 19-21. How much is the depth of interface of salt-fresh groundwater? And

I cannot find any reference to support "salt groundwater primarily occurring in the first aquifer of the delta area" in this part.

Response: We will introduce more detail of the location of salt-fresh interface, and add references (Li et al., 2013; Ma et al., 2014) to support "salt groundwater primarily occurring in the first aquifer of the delta area" in this part.

Page 10, line 2, remove"Holocene sea-land transition facies"and write "Holocene delta facies" .

Response: Change made.

Page 10, Line 1-3, the sedimetary phases in Fig. 2 should correspond to description of stratigraphic architecture.

Response: We will recheck and revise the text and Fig. 2.

Page 10, line 12, "7 ka BP". Abbreviations need to be checked, please check throughout the paper.

Response: We will recheck and revise abbreviations throughout the paper.

Page 12, line 5, "Daqingher". Do you mean Daqinghe?

Response: We are sorry for the negligence of writing. Change made.

Page 12, line 19, and Table 1, which water samples does "P18" represent? "CSW"? Please explain in the text.

Response: The "P18" represent "CSW", we will explain in the text.

Try to report Table 1 and 2 as online supplementary material.

Response: Table 1 and 2 will be reported as online supplementary material.

1a: The figure shows there are river samples "L01..." or "S01...", but I cannot find these samples in Table 1, please check.

Response: We will recheck and revise the sample label in Fig. 1a.

Some figure labels / legends are rather small and hard to read, e.g., figures 4, 6.

Response: The figures in paper will be further improve.

Page 26, Line 9-11, changing "palaeo-coast zone" to "palaeo-coast line" maybe more

appropriate. In addition, there need more international references to support "100 m depth below present sea level".

Response: Change made. Adding the references (Liu et al., 2020; Li et al., 2014) to support "100 m depth below present sea level".

Page 26, Line 2-7, there are summary of coastal brine groundwater formation, I agree with most of the interpretations. However, how about brine groundwater in other coastal aquifer? Whether these brines have similar formation processes? Appropriate extended discussion could further convey some new understanding that ideally is applicable to other study areas.

Response: The brine groundwater have been found in global coast areas such as Nile delta (van Engelen et al., 2019), Mediterranean (Antonellini et al., 2008; Sola et al., 2014) and Bohai Sea coast (Han et al., 2014; Li et al., 2017). Many researchers believe the hypersaline (or brine) groundwater are associated with fine sediments of barrier-lagoon environments during Middle Holocene (Giambastiani et al., 2013; Vallejos et al., 2018). We will improve this section to convey how the insights of this study improve understanding of brine groundwater in other coast areas.

Page 33, Line 18-19, "Laizhou Bay" is not covered in any part of the article, please delete.

Response: According to comments raised by Anonymous Referee #2, we decided to rewrite the last paragraph to highlight how our results contribute to the similar research in salinized coast aquifers, to increase the global relevance of the paper.

Reference:

Antonellini, M., Mollema, P., Giambastiani B., et al. 2008. Salt water intrusion in the coastal aquifer of the southern Po Plain, Italy. Hydrogeology Journal, (2008) 16:1541-1556.

Giambastiani, B., Colombani, N., Mastrocicco, M., et al., 2013. Characterization of the lowland coastal aquifer of Comacchio (Ferrara, Italy): Hydrology, hydrochemistry and evolution of the system. Journal of Hydrology, 501: 35-44.

Han, D. M., Song, X. F., Currell, M. J., et al., 2014. Chemical and isotopic constraints on the evolution of groundwater salinization in the coastal plain aquifer o fLaizhou Bay,China, Journal of Hydrology, 508, 12-27.

Li, J., Liang, X., Jin, M. G., et. al., 2012. Geochemical signature of aquitard pore water and its paleo-environment implications in Caofeidian Harbor, China. Geochemical Journal, 47, 37–50.

Li, J., Liang, X., Jin, M. G., et al., 2017. Origin and Evolution of Aquitard Porewater in the Western Coastal Plain of Bohai Bay, China. Groundwater, 55(6):917-925.

Li, G.X., Li, P., Liu, Y., et al., 2014. Sedimentary system response to the global sea level change in

the East China Seas since the last glacial maximum. Earth-Science Reviews, 139 (2014), 390–405.

Liu, J., Qiu, J., Saito, Y., et al., 2020. Formation of the Yangtze Shoal in response to the post-glacial transgression of the paleo-Yangtze (Changjiang) estuary, China. Marine Geology, 423(2020), 106080.

Ma, F. S., Wei, A. H., Deng, Q. H., et. al., 2014. Hydrochemical Characteristics and the Suitability of Groundwater in the Coastal Region of Tangshan, China. Journal of Earth Science, 26 (6), 1067–1075.

Vallejos, A., Sola, F., Yechieli, Y., Pulido-Bosch, A., 2018. Influence of the paleogeographic evolution on the groundwater salinity in a coastal aquifer. Cabo de Gata aquifer, SE Spain. Journal of Hydrology, 557 (2018) 55-66.

van Engelen, J., Verkaik, J., King, J., et al., 2019. A three-dimensional palaeohydrogeological reconstruction of the groundwater salinity distribution in the Nile Delta Aquifer. Hydrology and Earth System Sciences, 23, 5175-5198.

Sola, F., Vallejos, A., Daniele, L., Pulido-Bosch, A., 2014. Identification of a Holocene aquifer–lagoon system using hydrogeochemical data. Quaternary Research, 82 (2014) 121-131.

---

## Author Comment (AC2)

**Anonymous Referee #2:**

**General Comments**

This is potentially an interesting and valuable dataset; however, the paper as written does not do it justice. Most of the sections of the paper are too long and lack focus. The Introduction and Study Area sections are generally clearly written but it is not always clear how the information here relates to the groundwater chemistry (which is the main topic). They should be shorter and better focussed on the specifics of the work carried out.

Response: We will make substantial changes in response to this comment, and related subsequent comments (below), including a re-origanization of the paper to be shorten and highlight the key points. We will condense the Introduction and Study Area to fucus on background issues relevant to the topic.

Unfortunately, the Results and Discussion sections which are critical to the study are very hard to follow. Material is repeated, the writing is difficult to understand in places, and it is not clear what is important. The interpretation of the data (in particular the $^{14}$C) is superficial and uncritical. These sections really need rewriting.

Response: We will thoroughly revise the Results and Discussion to reduce repetition and give it a good readability. We will re-examine the interpretation of $^{14}$C data, and further revise the text according the related subsequent comments.

The Conclusions and Abstract also need to convey something of the general importance of this study and how it relates to work occurring elsewhere. Case studies are publishable in international journals such as HESS. However, unless they have relevance to researchers working elsewhere, they may be better in a regional journal.

Response: The Conclusions and Abstract will be re-organized and highlighting the relationship between researching elsewhere and the contribution made in this case. The relationship between this study and other research is as follows:

Variable groundwater types, including fresh, brackish, saline or brine, have been found in global coast areas (Larson et al., 2017), such as Nile delta (van Engelen et al., 2019), Mediterranean (Antonellini et al., 2008; Sola et al., 2014) and Bohai Sea coast (Liu et al., 2017; Li et al., 2017). Marine transgression deposits are often put forward to explain observed saline groundwater, while fresh and brackish are related to flushing during the

river deposits development (Post, 2004; Santucci et al., 2016; Han et al., 2020). Many researchers believe the hypersaline (or brine) groundwater are associated with fine sediments of barrier-lagoon environments during Middle Holocene (Giambastiani et al., 2013; Vallejos et al., 2018). Few previous studies examined cases involving multiple salinized processes and groundwater evolution throughout sedimentary deposition. Luanher River Delta (LRD) is a typical fan-shape delta developed from wave domination, which is similar to Nile delta. This study provided a novel case study, using a series of hydrochemical, isotopic and sedimentary indicators to identify the evolutionary pattern of saline groundwater and its link to LRD sedimentary setting. The insights of this study are also applicable to salinized aquifers throughout the world that have a similar sedimentary history, like Po River Delta in Italy (Colombani et al., 2017), Laizhou Bay in China (Han et al., 2014) and Western Port Bay in Australia (Lee et al., 2016).

Reference:

Antonellini, M., Mollema, P., Giambastiani B., et al. 2008. Salt water intrusion in the coastal aquifer of the southern Po Plain, Italy. Hydrogeology Journal, (2008) 16:1541-1556.

Caschetto, M., Colombani, N., Mastrocicco, M., et al.,2016. Estimating groundwater residence time and recharge patterns in a saline coastal aquifer. Hydrological Processes, 2016.

Colombani, N., Cuoco, E., Mastrocicco, M., 2017. Origin and pattern of salinization in the Holocene aquifer of the southern Po Delta (NE Italy). Journal of Geochemical Exploration, 175: 130-137.

Giambastiani, B., Colombani, N., Mastrocicco, M., et al., 2013. Characterization of the lowland coastal aquifer of Comacchio (Ferrara, Italy): Hydrology, hydrochemistry and evolution of the system. Journal of Hydrology, 501: 35-44.

Han, D. M., Song, X. F., Currell, M. J., et al., 2014. Chemical and isotopic constraints on the evolution of groundwater salinization in the coastal plain aquifer o fLaizhou Bay,China, Journal of Hydrology, 508, 12-27.

Larsen, F., Tran, L. V., Van Hoang, H., et. al., 2017. Groundwater salinity influenced by Holocene seawater trapped in incised valleys in the Red River delta. Nature Geoscience, 10, 376-381.

Lee, S., Currell, M., and Cendon, D. I., 2016. Marine water from mid-Holocene sea level highstand trapped in a coastal aquifer: Evidence from groundwater isotopes, and environmental significance. Science of the Total Environment, 544, 995-1007.

Li, J., liang, X., Jin, M. G., et al., 2017. Origin and Evolution of Aquitard Porewater in the Western Coastal Plain of Bohai Bay, China. Groundwater, 55(6):917-925.

Liu, S., Tang, Z., Gao, M. et. al., 2017. Evolutionary process of saline-water intrusion in Holocene and Late Pleistocene groundwater in southern Laizhou Bay. Science of The Total Environment, 607-608, 586-599.

Post V. E. A., 2004. Groundwater salinization processes in the coastal area of the Netherlands due to transgressions during the Holocene. Thesis, Vrije Universiteit, Amsterdam.

Vallejos, A., Sola, F., Yechieli, Y., Pulido-Bosch, A., 2018. Influence of the paleogeographic

evolution on the groundwater salinity in a coastal aquifer. Cabo de Gata aquifer, SE Spain. Journal of Hydrology, 557 (2018) 55-66.

van Engelen, J., Verkaik, J., King, J., et al., 2019. A three-dimensional palaeohydrogeological reconstruction of the groundwater salinity distribution in the Nile Delta Aquifer. Hydrology and Earth System Sciences, 23, 5175-5198.

Sola, F., Vallejos, A., Daniele, L., Pulido-Bosch, A., 2014. Identification of a Holocene aquifer–lagoon system using hydrogeochemical data. Quaternary Research, 82 (2014) 121-131.

**Specific Comments**

**Abstract**

The abstract is not clearly written and not that informative. For example, "The results of hydro-geochemical modeling (PHREEQC) suggest that the salty sources of salinization are seawater and concentrated saline water (formed after evaporation of seawater)" is not clear.

Response: We will recheck this section, and rephrased the sentences which are not clear.

There is also a lot of repetition: Page 2 lines 10-20 give the same information three times and some of the same information also appears on Page 3 lines 1 to 6.

Response: We will remove the repetition texts.

Try to put a bit more detail into the abstract (report the important results and highlight the important general points) rather than just the repeated brief summaries. Abstracts are important as they are what the reader uses to see if the paper might be worth reading, so they need to convey enough detail and a sense of importance.

Response: We will integrate more detail information about the important results and key points into the Abstract.

**Introduction**

The introduction covers a lot of topics, but it is not clear how the paper will address these topics. It has a general literature review feel to it rather than setting up the study. The final sentence seems to be indicating how prior research on sediment cores helps, which is not what the paper is about. Try to focus on aspects that relate more directly to the study and add an objectives section at the end so the reader has an idea of what you are trying to achieve.

Response: We will improve the Introduction to focus on background issue relevant to the topic, and add a paragraph to clearly introduce the objectives of this study.

Study area

This is comprehensive, but like much of the paper it is long. What details are important here and focus on those. Some of the geological history is a bit superfluous.

Response: We will remove the superfluous texts.

Results

The sections on major ion geochemistry (4.1) and stable isotopes (4.2) present the data but could be more succinct. There is a tendency to repeat information (especially in the major ion section).

Response: We will recheck the section 4.1 and 4.2, and remove the repeated information.

More importantly, there are some data that you interpret in Section 5 that would have been better presented here, for example you introduce Fig. 6 in section 5. If you are going to split the discussion from the results, make sure that you are not describing data in the discussion section.

Response: We will make change to ensure description of data is confined to Section 4.

Radiocarbon (Section 4.3)

This section deals with the data in a superficial way. Conventional radiocarbon ages assume simple one-dimensional, non-dispersive flow (piston flow) such that all the groundwater collected at the well was recharged at the same time. This is obviously an oversimplification as groundwater flows along paths of varying lengths and undergoes hydrodynamic dispersion and diffusion. Thus, groundwater has a range of residence times and, while a mean residence time may be defined, this does not equate to a specific age (Maloszewski and Zuber, 1982; Cook and Bohlke, 2000; Suckow, 2014). The use of a uniform input value for $^{14}$C of 100 pMC rather than accounting for the long term variation in atmospheric a$^{14}$C also yields "ages" in radiocarbon years (not ages BP as is in Table 2).
The combination of a variable atmospheric A$^{14}$C and more realistic flow models makes a non-trivial difference to calculated residence times of up to several thousand years in some cases (i.e. it is not just a matter of terminology: e.g., Cartwright et al., 2020). Additionally, many regional aquifers show macroscopic mixing between younger and older groundwater such that there are large volumes of groundwater that contain tritium but which also have "old" $^{14}$C (Jasechko, 2016; Jasechko et al., 2016). While you may

not have the data to assess some of these issues, you should at least acknowledge them and recognise the limitations. The correction for addition of $^{14}$C-free carbon from the aquifer matrix is not always correct. A simple way to check on the reasonableness of this calculation is to estimate what the initial A$^{14}$C of the Modern waters are. Those waters were recharged over the last few decades (post nuclear tests) so there has been negligible decay of $^{14}$C and the initial A$^{14}$C = measured $^{14}$C / q. The estimated initial A$^{14}$C values for the data in table 2 are: G01 = 125 pMC, G06 = 169 pMC, G07 = 104 pMC, G08 = 150 pMC, G09 = 139 pMC

The $^{14}$C activities in the atmosphere were as high as this following the nuclear tests but soil zone $CO_2$ (from where groundwater derives its DIC) are generally below 120 pMC (Jenkinson et al., 1992; Tipping et al., 2010) and I am not aware of modern groundwater with $^{14}$C activities any higher than that. Anomalously high estimates of initial A$^{14}$C (above 120 pMC) indicate that the correction cannot be correct. That is not necessarily surprising as the $^{13}$C of the end-members are not always well known and can be locally variable, and there are other unaccounted for processes (such as methanogenesis, open system calcite dissolution, recharge from river systems) that may be locally important. However, this needs to be recognised rather than just presenting the results uncritically.

Response: We are aware of that interpretation of $^{14}$C data define a mean residence time of groundwater, instead of specific age. We are sorry for the unprecise expression in this section. According the above comments, we recognized that some factors including possible factors that may influence the $^{14}$C value and, limitations of the correction should be taken into considerations. To interpret the $^{14}$C data properly, the improvements of this section would be done.

The distribution of $^{14}$C activities with depth implies that the general interpretation here is correct; however, the details of the interpretation are oversimplified; at the very least some error propagation is needed.

Response: We will conduct more uncertainty analysis about groundwater residence time in this part.

Discussion

This is not very well written and it loses focus. I generally agree with the results but the explanations tend to be overly long and very confused.

Response: A full-reorganisation of this section will be conducted to ensure most of the

introductory material is significantly condensed and consolidated.

Section 5.1

The relative residence times here are fine; however, this section needs to deemphasise the discussion of absolute ages (see above).

Response: We will make change to avoid the discussion of absolute ageds.

Some of the terminology is poor ("has a slightly higher stable isotope content than deeper groundwater, which is typical of the recharge source as the atmosphere has changed since the last deglaciation") – I can guess what this means but it is verging on being unintelligible.

Response: The sentences will be rephrased.

Some of the material here is repeated later – for example you discuss mixing at the bottom of page 20, but that is repeated in Section 5.3

Response: We will merge the repeated part into Section 5.3.

Section 5.2.

I am not sure what the Scholler plot adds. It is a common observation that saline groundwater has a similar geochemistry to ocean water (not because it is always necessarily derived from ocean water as mineral precipitation and ion exchange can modify its geochemistry during evaporation). You have reported the salinities and water types, which is enough.

Response: We will remove the interpretation of Scholler plot to avoid superfluous discussion.

Here again, the explanation of the results is not always clear (e.g., "the salinity of salinization groundwater mainly originates from seawater or, the CSW which is subject to evaporated seawater" and "Due to reach saturation, there were loss of ions follow mineral precipitation such as…" and "Calcite and gypsum will be dissolved along with surface water during lateral recharge, resulting in brackish groundwater samples plotted above the mixing line, highlighting surface water flushing processes in the study region"). Having to guess the meaning of these sentences detracts from the study.

Response: We will rephrase the sentences to make sure that the meaning is clear.

It is not always clear what the important points are here, so while you are probably interpreting the processes correctly, why are they important? Somewhere in this section, you need to explain how this information relates to your overall objectives and why these pieces of information are important.

Response: This section will be further improved to clearly explain the implication of salty sources and hydrochemical evolution for salinization processes.

Section 5.3.

The general model of mixing (Fig. 9) is also probably correct and it is clearer from the objectives why you are doing this. However, again this section could be shorter; the general introduction on the first few lines is probably not needed and the explanations on Pag 25 are repetitious. As with the rest of the discussion section, there are no attempts to justify the results (the end-members for example are just assigned without comment).

Response: According the comments, we will remove the superfluous part, and further discuss the reasonability of end-members in different mixing processes.

Section 6

This is far better written than most of the paper. It is still long and some of the narrative could be shorter. This material is not generally well linked to the geochemistry and it is not always clear how much it is a synthesis of previous studies rather than a discussion of this study.

Response: The texts will be shortened, and revised to clearly delineates the roles of hydrochemistry in evolutionary pattern of groundwater.

Conclusions

Most of these repeat details from the main part of the study. It would be better with a much briefer summary of these and some consideration of how what you have done here has improved understanding of processes in these environments more generally. Also, how do your results fit into the broader research going on elsewhere. Explaining that will give the paper more impact.

The last paragraph does not relate well the to study as there is no discussion of groundwater levels, monitoring, or policy. While those things may be important, it is

not clear how your research informs them. Perhaps that could be the focus of this section?

Response: According to comments, we will re-organize the Conclusions, and rewrite the last paragraph to highlight how the insights of this study improve understanding of groundwater salinization elsewhere, to increase the global relevance of the paper.

---

## Author Response (AR1)

**Responses for reviewers**

Oct 23, 2021

⁓⁓⁓⁓⁓⁓⁓⁓⁓⁓⁓⁓⁓⁓⁓⁓⁓⁓⁓⁓⁓⁓⁓⁓⁓⁓⁓⁓⁓⁓⁓⁓⁓⁓⁓⁓⁓⁓⁓⁓⁓⁓⁓⁓⁓⁓⁓⁓⁓⁓⁓⁓⁓⁓⁓⁓⁓⁓⁓⁓⁓⁓⁓⁓⁓⁓⁓

**Manuscript Number:** hess-2021-246

**Manuscript Title:** Saline groundwater evolution in Luanhe River Delta, China since Holocene: hydrochemical, isotopic and sedimentary evidence

**Authors:** Xianzhang Dang, Maosheng Gao, Zhang Wen, Guohua Hou, Hamza Jakada, Daniel Ayejoto, Qiming Sun

⁓⁓⁓⁓⁓⁓⁓⁓⁓⁓⁓⁓⁓⁓⁓⁓⁓⁓⁓⁓⁓⁓⁓⁓⁓⁓⁓⁓⁓⁓⁓⁓⁓⁓⁓⁓⁓⁓⁓⁓⁓⁓⁓⁓⁓⁓⁓⁓⁓⁓⁓⁓⁓⁓⁓⁓⁓⁓⁓⁓⁓⁓⁓⁓⁓⁓⁓

**Responses for reviewers#1**

Thank you for your comments concerning our manuscript, we found these comments will help to improve its quality greatly. We have attempted to address each of the comments point-by-point. Detail explanations can be found as follows.

Author's response –Line numbers referring to the old and new version manuscripts are preceded by L and RL, respectively

**General Comment:**

The authors focus on observed saline groundwater that extends far inland in the coastal zones, and investigate the origin of groundwater salinity and elucidate the major processes controlling salinized groundwater evolution in Luanhe River Delta using hydrochemical and isotopic methods and sedimentary characteristics. The method for interpreting the data is relatively standard. The text is well structured and well supported by the figures. The evolutionary pattern of saline groundwater is innovative, and can be used to better understand groundwater evolution in coastal zone. I would like to recommend the acceptance of the manuscript for publication after minor revisions. Below are my comments that may help with this process.

**Response:** Thank you for your general comments concerning our manuscript. The manuscript has been improved and polished carefully for readability and succinctness.

We have responded to the following specific comments in depth.

**Specific comments:**

The introduction provides a comprehensive summary of the groundwater salinization in coastal zones due to Quaternary transgression. The relevant study of similar coastal zones could be expended a little as it is contributed to more valuable scientific significance in International Journals.

**Response:** We have revised the last paragraph of "Introduction" to introduce previous study in Bohai Sea coast, as well as highlight the novel contribution made in the current work.

*P5 RL8 to 22:* "The Bohai Sea of northern China was affected by Late Pleistocene transgressive-regressive cycles, which caused various salinity palaeo-saltwater intrusion along the coastal aquifers (Du et al., 2015; Li et al., 2017). Several studies have applied geochemical methods to elucidate the origin of saline groundwater and the salinization processes under anthropogenic influence, including induced mixing brine water from adjacent aquifers caused by groundwater overexploitation in Laizhou Bay (Han et al., 2011, 2014; Liu et al., 2017; Qi et al., 2019). However, the association between groundwater salinization (especially brine formation) and palaeoenvironmental implications are still not clear. Thus, this study applies a range of chemical, isotopic and sedimentary indicators to examine the Luanhe River Delta (situated along the northwestern coast of Bohai Sea) to elucidate the groundwater salinization processes in relation to recharge, salt source, mixing behavior and palaeogeographic evolution. The overall goal is to understand the groundwater evolutionary pattern influenced by transgression/regression events in geologic time. The findings will be significant to aquifer remediation activities in the region as well as other similar sedimentary environments around the world."

Hydrogeology: this paper is aimed to saline groundwater in coastal aquifer of Luanhe River Delta. However, I cannot find more information about water quality in this section. And how about the hydraulic connection between shallow and deep aquifers? The authors should introduce more details here.

**Response:** We have added the information on groundwater quality in P6 RL16 to17, and further discuss the hydraulic connection between shallow and deep aquifers P7 RL3 to 4..

Page 9, line 6, "MIS5". The first abbreviations need to be explained in detail.

**Response:** The first abbreviations have been re-check, and explained clearly.

*P11 RL8:* "radiocarbon ages in years Before Present (a B.P.)"

*P23 RL4:* "Marine isotope stage (MIS)"

Page 9, lines 19-21. How much is the depth of interface of salt-fresh groundwater? And I cannot find any reference to support "salt groundwater primarily occurring in the first aquifer of the delta area" in this part.

**Response:** We have added the location information of salt-fresh interface in P8 RL9, and references (Li et al., 2013; Ma et al., 2014) to support "salt groundwater primarily occurring in the first aquifer of the delta area" in this part.

Page 10, line 2, remove "Holocene sea-land transition facies" and write "Holocene delta facies".

Response: Change made. P8 RL13 "Holocene delta facies".

Page 10, Line 1-3, the sedimetary phases in Fig. 2 should correspond to description of stratigraphic architecture.

**Response:** We have rechecked and revised the text and Fig. 2.

[Figure]

Fig.2 Stratigraphic transect along the present coastline of Luanhe River Delta, modified from He et al.,2020.

Page 10, line 12, "7 ka BP". Abbreviations need to be checked, please check throughout the paper.

**Response:** "7 ka BP" change to "7 ka B.P." in P9 RL1.

Page 12, line 5, "Daqingher". Do you mean Daqinghe?

**Response:** We are sorry for the negligence of writing. Change made.

Page 12, line 19, and Table 1, which water samples does "P18" represent? "CSW"? Please explain in the text.

**Response:** The "P18" represent "CSW", P10 RL5 "1CSW (P18 sample)".

Try to report Table 1 and 2 as online supplementary material.

**Response:** Table 1 and 2 will be reported as online supplementary material (Table S1 and S2).

1a: The figure shows there are river samples "L01..." or "S01...", but I cannot find these samples in Table 1, please check.

**Response:** We have revise the sample label in Fig. 1a.

[Figure]

Some figure labels / legends are rather small and hard to read, e.g., figures 4, 6.

**Response:** The Fig 4 has been improved, while the Fig 6. has been removed to avoid superfluous discussion (as suggested by reviewer#2).

[Figure]

Fig. 4 Stable isotope compositions of different water samples. Seawater mixing line: mixing between deep fresh groundwater and seawater; CSW mixing line: mixing between deep fresh groundwater and CSW.

Page 26, Line 9-11, changing "palaeo-coast zone" to "palaeo-coast line" maybe more appropriate. In addition, there need more international references to support "100 m depth below present sea level".

**Response:** Change made, P23 RL8 "palaeo-coast line", and we have added the references (Liu et al., 2020; Li et al., 2014) in P23 RL9 to support "100 m depth below present sea level".

Page 26, Line 2-7, there are summary of coastal brine groundwater formation, I agree with most of the interpretations. However, how about brine groundwater in other coastal aquifer? Whether these brines have similar formation processes? Appropriate extended discussion could further convey some new understanding that ideally is applicable to other study areas.

**Response:** We have improved this section to convey how the insights of this study improve understanding of brine groundwater in Bohai sea coast.

*P27 RL12 to 19:* "The coastal brine groundwater is a special product of geological evolution, which have been found in Bohai Sea coast such as Bohai Bay (Li et al., 2017) and Laizhou Bay (Han et al., 2014). The change in sea level over the Late Pleistocene would have favoured marine intrusion and similar sedimentary environment in Bohai coast, allowing this study infers the following conditions for its brine formation: (1) stable evaporative environments (e.g. lagoon), (2) suitable climatic conditions (e.g. arid), (3) seawater entering evaporative environments (e.g. storm or tide), and (4) long-term scale for salinity accumulation."

Page 33, Line 18-19, "Laizhou Bay" is not covered in any part of the article, please delete.

Response: According to comments raised by reviewer #2, we have rewritten the last paragraph to highlight how our results contribute to the similar research in salinized coast aquifers, to increase the global relevance of the paper.

*P29 RL20 to P30 RL7:* "Given that most coastal zones around the world experienced transgression/regression events in the Quaternary period, the findings of this work will promote better understanding of the origin of salinization in coastal aquifers. In addition, it is important to recognize the potential leak of connate saline groundwater previously preserved in adjunct aquifers that can occur due to over-extraction of deep groundwater. To effectively prevent pollution from saline groundwater movement, this study recommends extensive characterization of groundwater interface dynamics, such as fresh/saline, fresh/brine, and brine/seawater interfaces and also maintain continuous monitoring of water quality and levels across the aquifers."

**Responses for reviewers#2**

Thank you for your comments on our manuscript. We found the comments helpful towards improving the quality of the manuscript. We have attempted to address each of the comments point-by-point. Detail explanations are as follows.

Author's response –Line numbers referring to the old and new version manuscripts are preceded by L and RL, respectively

**General Comments**

This is potentially an interesting and valuable dataset; however, the paper as written does not do it justice. Most of the sections of the paper are too long and lack focus. The Introduction and Study Area sections are generally clearly written but it is not always clear how the information here relates to the groundwater chemistry (which is the main topic). They should be shorter and better focussed on the specifics of the work carried out.

**Response:** Thank you for the positive comments. We have reorganized the paper in response to your recommendations, including condensing the introductory material highlighting the key points. We have removed section 2.2 and some negligible information in section 1 which are not very relevant to the topic of this work. We also changed "P5 L11 to P6 L5" to "P5 RL8 to 22", showing clear background issues relevant to the topic.

*P5 RL8 to 22:* "The Bohai Sea of northern China was affected by Late Pleistocene transgressive-regressive cycles, which caused various salinity palaeo-saltwater intrusion along the coastal aquifers (Du et al., 2015; Li et al., 2017). Several studies have applied geochemical methods to elucidate the origin of saline groundwater and the salinization processes under anthropogenic influence, including induced mixing brine water from adjacent aquifers caused by groundwater overexploitation in Laizhou Bay (Han et al., 2011, 2014; Liu et al., 2017; Qi et al., 2019). However, the association between groundwater salinization (especially brine formation) and

palaeoenvironmental implications are still not clear. Thus, this study applies a range of chemical, isotopic and sedimentary indicators to examine the Luanhe River Delta (situated along the northwestern coast of Bohai Sea) to elucidate the groundwater salinization processes in relation to recharge, salt source, mixing behavior and palaeogeographic evolution. The overall goal is to understand the groundwater evolutionary pattern influenced by transgression/regression events in geologic time. The findings will be significant to aquifer remediation activities in the region as well as other similar sedimentary environments around the world."

Unfortunately, the Results and Discussion sections which are critical to the study are very hard to follow. Material is repeated, the writing is difficult to understand in places, and it is not clear what is important. The interpretation of the data (in particular the 14C) is superficial and uncritical. These sections really need rewriting.

**Response:** We have thoroughly revised the Results and Discussion to reduce repetition and improve the readability. We have re-examined the interpretation of $^{14}$C data (revised section 4.3), and further revised the text to also improve the discussion.

The Conclusions and Abstract also need to convey something of the general importance of this study and how it relates to work occurring elsewhere. Case studies are publishable in international journals such as HESS. However, unless they have relevance to researchers working elsewhere, they may be better in a regional journal.

**Response:** Thank you for this important suggestion. We have revised the Conclusions and Abstract to convey the major significance of our work for the scientific community working elsewhere.

*P3 RL3 to 5:* "This study presents an approach for utilizing geochemical indicator analysis with paleogeographic reconstruction to better assess groundwater evolutionary patterns in coastal aquifers."

*P29 RL11 toP30 RL 7:* "Our study shows that multiple water types are particularly associated with complex geographic evolution in coastal areas. The variation in sea-levels (when it rises) causes lowland coastal areas to be inundated by seawater, which induces palaeo-seawater intrusion. The costal deltas developed after significant drop in the sea levels. The concentration of saline water in the lagoon environment at the deltafront continuously provided salinity to the groundwater. Thus, under the effects of evaporation, mixing, and dissolution, brine groundwater was formed. In contrast, the lateral recharge of surface water and irrigation return would cause slow wash-out of salinized groundwater in the delta plain.

Given that most coastal zones around the world experienced transgression/regression events in the Quaternary period, the findings of this work will promote better understanding of the origin of salinization in coastal aquifers. In addition, it is important to recognize the potential leak of connate saline groundwater previously preserved in adjunct aquifers that can occur due to over-extraction of deep groundwater. To effectively prevent pollution from saline groundwater movement, this study recommends extensive characterization of groundwater interface dynamics, such as fresh/saline, fresh/brine, and brine/seawater interfaces and also maintain continuous monitoring of water quality and levels across the aquifers."

**Specific Comments**

Abstract

The abstract is not clearly written and not that informative. For example, "The results of hydro-geochemical modeling (PHREEQC) suggest that the salty sources of salinization are seawater and concentrated saline water (formed after evaporation of seawater)" is not clear.

There is also a lot of repetition: Page 2 lines 10-20 give the same information three times and some of the same information also appears on Page 3 lines 1 to 6.

Try to put a bit more detail into the abstract (report the important results and highlight the important general points) rather than just the repeated brief summaries. Abstracts are important as they are what the reader uses to see if the paper might be worth reading, so they need to convey enough detail and a sense of importance.

**Response:** We have rewritten the unclear sections and removed the repetitive texts. Also, the main results and key points have been incorporated into the Abstract.

*P2 RL10 to P3 RL2:* "The isotopic indicators ($^2$H, $^{18}$O, $^{14}$C) facilitate the distinciton between old and new groundwater recharge. Results of the hydro-chemical analysis using PHREEQC indicate that the origin of salt in saline and brine groundwater is from a marine source. The $^{18}$O-Cl relationship diagram yields three end-member mixing of groundwater with wo mixing scenarios suggested to explain the freshening and

salinization processes in study area. When interpreted with data from palaeo-environmental sediments, we found that groundwater salinization may have occurred since the Holocene marine transgression. The brine is characterized by radiocarbon activities of ~50 to 85 pMC and relatively depleted stable isotopes, which is associated with seawater evaporation in the ancient lagoon during delta progradation, as well as mixing with deeper fresh groundwater which probably was recharged in cold late Pleistocene. As for the brackish and fresh groundwater, they are characterized by river-like stable isotope values where high radiocarbon activities (74.3 to 105.9 pMC) were formed after the wash-out of salinized aquifer by surface water in the delta plain."

Introduction

The introduction covers a lot of topics, but it is not clear how the paper will address these topics. It has a general literature review feel to it rather than setting up the study. The final sentence seems to be indicating how prior research on sediment cores helps, which is not what the paper is about. Try to focus on aspects that relate more directly to the study and add an objectives section at the end so the reader has an idea of what you are trying to achieve.

**Response:** As shown above, we have rewritten the last paragraph of Introduction, and added a paragraph to clearly introduce the objectives of this study. Some irrelevant sections have also have been removed.

Study area

This is comprehensive, but like much of the paper it is long. What details are important here and focus on those. Some of the geological history is a bit superfluous.

**Response:** "MIS 3 and 5 transgressions" information in the previous section 2.2 is not very relevant. The narrative on the "Holocene transgression" has been incorporated into previous section 2.3, whereas section 2.2 has been removed.

Results

The sections on major ion geochemistry (4.1) and stable isotopes (4.2) present the data but could be more succinct. There is a tendency to repeat information (especially in the major ion section).

**Response:** We have removed the repeated information, and consolidated the narratives to make it more succinct (revised 4.1 and 4.2).

More importantly, there are some data that you interpret in Section 5 that would have been better presented here, for example you introduce Fig. 6 in section 5. If you are going to split the discussion from the results, make sure that you are not describing data in the discussion section.

**Response:** We have made changes to ensure description of data is confined to Section 4. The previous Fig. 6 and its narrative have been removed (as suggested by reviewer).

Radiocarbon (Section 4.3)

This section deals with the data in a superficial way. Conventional radiocarbon ages assume simple one-dimensional, non-dispersive flow (piston flow) such that all the groundwater collected at the well was recharged at the same time. This is obviously an oversimplification as groundwater flows along paths of varying lengths and undergoes hydrodynamic dispersion and diffusion. Thus, groundwater has a range of residence times and, while a mean residence time may be defined, this does not equate to a specific age (Maloszewski and Zuber, 1982; Cook and Bohlke, 2000; Suckow, 2014). The use of a uniform input value for 14C of 100 pMC rather than accounting for the long term variation in atmospheric a14C also yields "ages" in radiocarbon years (not ages BP as is in Table 2).

The combination of a variable atmospheric A14C and more realistic flow models makes a non-trivial difference to calculated residence times of up to several thousand years in some cases (i.e. it is not just a matter of terminology: e.g., Cartwright et al., 2020).

Additionally, many regional aquifers show macroscopic mixing between younger and older groundwater such that there are large volumes of groundwater that contain tritium but which also have "old" 14C (Jasechko, 2016; Jasechko et al., 2016). While you may not have the data to assess some of these issues, you should at least acknowledge them and recognise the limitations. The correction for addition of 14C-free carbon from the aquifer matrix is not always correct. A simple way to check on the reasonableness of this calculation is to estimate what the initial A14C of the Modern waters are. Those waters were recharged over the last few decades (post nuclear tests) so there has been negligible decay of 14C and the initial A14C = measured 14C / q. The estimated initial A14C values for the data in table 2 are: G01 = 125 pMC, G06 = 169 pMC, G07 = 104 pMC, G08 = 150 pMC, G09 = 139 pMC

The 14C activities in the atmosphere were as high as this following the nuclear tests but

soil zone CO2 (from where groundwater derives its DIC) are generally below 120 pMC (Jenkinson et al., 1992; Tipping et al., 2010) and I am not aware of modern groundwater with 14C activities any higher than that. Anomalously high estimates of initial A14C (above 120 pMC) indicate that the correction cannot be correct. That is not necessarily surprising as the 13C of the end-members are not always well known and can be locally variable, and there are other unaccounted for processes (such as methanogenesis, open system calcite dissolution, recharge from river systems) that may be locally important. However, this needs to be recognised rather than just presenting the results uncritically. The distribution of 14C activities with depth implies that the general interpretation here is correct; however, the details of the interpretation are oversimplified; at the very least some error propagation is needed.

**Response:** According the Reviewer's comments, significant changes have been made in the section cited. We have revised the section to introduce factors that may influence the $^{14}$C value and limitations.

*P14 RL10 to P16 RL2:* "The properties of $^{14}$C and sampling depth is shown in Fig. 5, which elucidates the negative correlations , showing that variations of $^{14}$C activities could be attributed to radioactive decay aquifer. There are multiple processes that can impact the $^{14}$C properties including groundwater mixing and dispersion, long-term variation of atmospheric $^{14}$C and free $^{14}$C dilution (e.g. carbonate dissolution) (Cartwright et al., 2020). Due to the relative impact of these processes (which are not well established in the study area), the uncertainty regarding the correction of radiocarbon ages to real groundwater ages is very high. Consequently, we estimate groundwater age as a range of the residence time. Uncorrected ages are considered the maximum age, while corrected ages are the minimum age that are determined based on two hypothetical models on carbonate dissolution that mainly affect the $^{14}$C contents of water samples (Lee et al., 2016).

Fig.5 shows activities of the $^{14}$C in the shallow groundwater are within 30.6 to105.9 pMC. These values indicate relatively modern recharge before atmospheric nuclear testing period of the 1950s and 1960s. The radiocarbon activities in the deep fresh groundwater are less than 12 pMC, which is consistent with the palaeo-water recharge. This indicates that there are weak connection between shallow and deep aquifers. Therefore, we assume that the shallow aquifer is an open system, while the deep aquifer is a closed system. The $\delta^{13}$C mixing and chemical mass balance (CMB) models are used

to estimate to corrective factor q, respectively (Clark and Fritz, 1997).

For $\delta^{13}$C mixing model, $q = (\delta^{13}C_{DIC} - \delta^{13}C_{CARB})/(\delta^{13}C_{RECH} - \delta^{13}C_{CARB})$ (Pearson and Hanshaw, 1970), where $\delta^{13}C_{DIC}$ is the measured $\delta^{13}$C of DIC in groundwater; $\delta^{13}C_{CARB}$ is the $\delta^{13}$C of DIC from dissolved soil mineral, using $\delta^{13}C_{CARB} = 1.5$ ‰ (Chen et al., 2003); $\delta^{13}C_{RECH}$ is the $\delta^{13}$C in water when it reaches the saturation zone. In this study, we use a $\delta^{13}C_{RECH}$ of -15 ‰, which has been suggested as appropriate for soils in northern China dominated by $C_4$ plants (Currell et al., 2010). The model yielded some relatively low q values (0.59 of G06-15 and 0.65 of G08-15), possibly since several unaccounted factors would contribute to variable $\delta^{13}C_{RECH}$ values, e.g. local methanogenesis and pH or temperatures in the soil zones."

Discussion

This is not very well written and it loses focus. I generally agree with the results but the explanations tend to be overly long and very confused.

**Response:** The entire section has been reorganized to ensure succinctness.

Section 5.1

The relative residence times here are fine; however, this section needs to deemphasise the discussion of absolute ages (see above).

**Response:** We have made change to avoid the discussion of absolute ages, such as P17 RL 11 to 12 "residence time of P15-150 and P14-300 samples (range from 33951 to 39050 a B.P)".

Some of the terminology is poor ("has a slightly higher stable isotope content than deeper groundwater, which is typical of the recharge source as the atmosphere has changed since the last deglaciation") – I can guess what this means but it is verging on being unintelligible.

**Response:** The sentences have been rephrased.

*P17 RL 13 to 14:* "The stable isotopes of P16-100 are more enriched, reflecting recharge history of warm climate during the last deglaciation."

Some of the material here is repeated later – for example you discuss mixing at the bottom of page 20, but that is repeated in Section 5.3

**Response:** We have merged and consolidated the repeated part (P19 L3 to 11) into Section 5.3.

Section 5.2.

I am not sure what the Scholler plot adds. It is a common observation that saline groundwater has a similar geochemistry to ocean water (not because it is always necessarily derived from ocean water as mineral precipitation and ion exchange can modify its geochemistry during evaporation). You have reported the salinities and water types, which is enough.

**Response:** We have removed the interpretation of Scholler plot to avoid superfluous discussion.

Here again, the explanation of the results is not always clear (e.g., "the salinity of salinization groundwater mainly originates from seawater or, the CSW which is subject to evaporated seawater" and "Due to reach saturation, there were loss of ions follow mineral precipitation such as…" and "Calcite and gypsum will be dissolved along with surface water during lateral recharge, resulting in brackish groundwater samples plotted above the mixing line, highlighting surface water flushing processes in the study region"). Having to guess the meaning of these sentences detracts from the study.

**Response:** The sentences have been rephrased to make sure that the meaning is clear.

*P18 RL19:* "the salt in these water samples is mainly of marine origin."

*P19 RL3 to 10:* "(1) $Ca^{2+}$ depletion of P18 and P12 samples are shown in Fig. 6d. This phenomenon is likely explained by gypsum ($CaSO_4$) precipitation. The evaporation line reveals that the $Ca^{2+}$ composition of evaporating seawater follows a hooked trajectory (Fig. 6d). During evaporation to the point of gypsum saturation, residual CSW becomes progressively decreased $Ca^{2+}$ concentration. (2) $Ca^{2+}$ and $SO_4^{2-}$ excess in most fresh and brackish samples (Fig. 6c and d) could be attributed to mineral dissolution along with stream water recharging, highlighting some degree of dilution with continental runoff since Holocene regression."

It is not always clear what the important points are here, so while you are probably interpreting the processes correctly, why are they important? Somewhere in this section,

you need to explain how this information relates to your overall objectives and why these pieces of information are important.

**Response:** We have further improved this section to clearly explain the implication of salty sources and hydrochemical evolution for salinization and/or freshening processes.

*P18 RL20 to 22:* "The major ions concentration in some samples (such as brine) are higher than those in the seawater, suggesting the enriched ions are associated with evaporation processes, rather than seawater intrusion (Colombani et al, 2017)."

*P19 RL8 to 10:* "$Ca^{2+}$ and $SO_4^{2-}$ excess in most fresh and brackish samples (Fig. 6c and d) could be attributed to mineral dissolution along with stream water recharging, this may indicate that some degree of dilution with continental runoff occurred since Holocene regression."

Section 5.3.

The general model of mixing (Fig. 9) is also probably correct and it is clearer from the objectives why you are doing this. However, again this section could be shorter; the general introduction on the first few lines is probably not needed and the explanations on Pag 25 are repetitious. As with the rest of the discussion section, there are no attempts to justify the results (the end-members for example are just assigned without comment).

**Response:** According the comments, we have removed the superfluous part, and further discussed the reasonability of end-members in different mixing processes.

*P21 RL8 to 14:* "Stable isotopes of high TDS saline and brine samples fall between the seawater and CWS mixing lines, further suggesting potential three end-member mixing processes (Douglas et al., 2000). Therefore, we considered SW01 (seawater but with most enriched $\delta^{18}O$) and P18 (most saline but with relatively depleted $\delta^{18}O$) as two saline end-members. The P16-100, which is most likely recharged during the Last Deglaciation, was chosen to represent fresh end-members that could have been impacted by overlying seawater or CSW during Holocene transgression."

*P22 RL4 to 7:* "The LH02 (depleted $\delta^{18}O$) and SH02 (relatively enriched $\delta^{18}O$) were selected to represent river water end-members range for different continental runoff in

study area, while the G09-15 (saline but with river-like stable isotope) was considered as a groundwater end-member."

Section 6

This is far better written than most of the paper. It is still long and some of the narrative could be shorter. This material is not generally well linked to the geochemistry and it is not always clear how much it is a synthesis of previous studies rather than a discussion of this study.

**Response:** We have removed the superfluous discussion and focused on delineating the link between groundwater evolution and palaeoenvironmental settings.

*P25 RL7 to 14:* "The good fit between the measured hydrochemistry and simulated evaporation lines (Fig. 6 and 7) is an indicator that the brine samples were associated with the seawater which was exposed to evaporation during geological history. Previous research has revealed that lagoon was active during the progradation of the old Luanhe River Delta between 7 and 3.5 ka B.P. (He et al., 2020; Xu et al., 2020). Meanwhile, the relatively arid climate had been developed since 5500 a B.P., which may lead to increased evaporation (Jin, 1984). The ancient lagoon would be an ideal location for evaporating seawater that had been trapped due to storms or tides (Fig. 9b)."

*P26 RL18 to P26 RL4:* "In addition, the brackish and low TDS saline groundwater with relatively modern age (e.g. G09-15), and river-like stable isotopes (Fig. 4 and 8), are compelling evidence that freshening processes have occurred in the delta plain. Since the semi-humid palaeoclimate, some abandoned channels have developed into small rivers after the diversion of the ancient Luanhe River (Gao, 1981), such as the Suhe River and Shahe River. Firstly, the lateral recharge from the surface stream plays a role in washing out the salty groundwater. Secondly, due to the inefficiency of saline groundwater throughout human history, river irrigation has been commonly used for agricultural activities in the study region, freshening the upper saline aquifer (Fig. 9c)."

Conclusions

Most of these repeat details from the main part of the study. It would be better with a much briefer summary of these and some consideration of how what you have done here has improved understanding of processes in these environments more generally.

Also, how do your results fit into the broader research going on elsewhere. Explaining that will give the paper more impact.

The last paragraph does not relate well the to study as there is no discussion of groundwater levels, monitoring, or policy. While those things may be important, it is not clear how your research informs them. Perhaps that could be the focus of this section?

**Response:** According to comments, we have re-organized the Conclusions, and rewritten the last paragraph to highlight how the insights of this study improve understanding and management of coastal groundwater systems elsewhere.

*P29 RL 11 to P30 RL7:* "Our study shows that multiple water types are particularly associated with complex geographic evolution in coastal areas. The variation in sea-levels (when it rises) causes lowland coastal areas to be inundated by seawater, which induces palaeo-seawater intrusion. The costal deltas developed after significant drop in the sea levels. The concentration of saline water in the lagoon environment at the delta-front continuously provided salinity to the groundwater. Thus, under the effects of evaporation, mixing, and dissolution, brine groundwater was formed. In contrast, the lateral recharge of surface water and irrigation return would cause slow wash-out of salinized groundwater in the delta plain.

Given that most coastal zones around the world experienced transgression/regression events in the Quaternary period, the findings of this work will promote better understanding of the origin of salinization in coastal aquifers. In addition, it is important to recognize the potential leak of connate saline groundwater previously preserved in adjunct aquifers that can occur due to over-extraction of deep groundwater. To effectively prevent pollution from saline groundwater movement, this study recommends extensive characterization of groundwater interface dynamics, such as fresh/saline, fresh/brine, and brine/seawater interfaces and also maintain continuous monitoring of water quality and levels across the aquifers."

---

## Author Response (AR2)

**Responses for reviewers#3**

Jan 26, 2022

~~

**Manuscript Number:** hess-2021-246

**Manuscript Title:** Saline groundwater evolution in Luanhe River Delta, China since Holocene: hydrochemical, isotopic and sedimentary evidence

**Authors:** Xianzhang Dang, Maosheng Gao, Zhang Wen, Guohua Hou, Hamza Jakada, Daniel Ayejoto, Qiming Sun

~~

Thank you for your comments concerning our manuscript, we found these comments will help to improve its quality greatly. We have attempted to address each of the comments point-by-point. Detail explanations are as follows.

Author's response –Line numbers referring to the old and revised version manuscripts are preceded by L and RL, respectively

**General Comment:**

This manuscript presents valuable data that should be published.

However, the presentation is terrible, compounded by very poor English with many incomprehensible sentences and sentences without verbs. Further, I am taken aback by clearly incorrect statements such as claiming that groundwater is the primary source of fresh water in coastal areas. The statement that 20 - 40% of the world's population lives in coastal areas is also not very accurate as most recent studies give a value of around 40%.

**Response:** Thank you for your general comments and valuable reminder. We are sorry for some unclear sentences, the manuscript has been further improved and polished carefully for readability and English language. We have changed "P4 L2 to 4" to "P4

RL2 to 5", showing more accurate statements.

*P4 RL2 to 5*: "It is estimated that around 40% of the world's population lives in coastal areas. (UN Atlas, 2010). Groundwater is the important freshwater resource for domestic consumption and agricultural activities in this region (Cary et al., 2015; Jayathunga et al., 2020)."

**Specific Comments**

1. It defines the brackish water as having a TDS between 1 and 3 g/L. In my field, it is up to 30 g/L.

**Response:** In coastal areas, water types in aquifers are often complex with varying characteristics. Similarly, there are diverse ways to classify these water types. In this study, groundwater samples were observed to have a wide range of TDS between 0.38-125.9 g/L. Therefore, we classified the groundwater into four types on the basis of TDS: the fresh water (TDS is less than 1 g/L), the brackish water (TDS between 1 and 3 g/L, representing salt water with relative low salinity), the saline water (TDS between 3 and 50 g/L, representing salt water with relative high salinity) and brine water (TDS is higher than 50 g/L).

2. Again, in my field it describes isotope values as heavier or lighter, not higher or lower.

**Response:** We are sorry for the inaccurate statements, we have rechecked and revised the text thoroughly.

3. It is not clear how DIC could be estimated based on pH and T. At least one more carbonate species is needed.

**Response:** Thank you for the comments, we have changed "P16 L7 to 9" to "P16 RL7 to 11".

*P16 RL7 to 11*: "$DIC_{rech}$ was mainly $HCO_3$ in recharge water when pH value was between 6.4 and 10.3, and the carbonate equilibrium constant varies with temperature (Clark and Fritz, 1997). $mDIC_{rech}$ was calculated from estimated pH and temperature conditions for the recharge environment, e.g., at pH = 6 and T = 15°C, the $mDIC_{rech}$=10 mmol/L (Currell et al., 2010)."

4. It is not clear what it means by " stable isotopes are more enriched"?. H-1, H-2, O-18,

O-16, C-12, and C-13 are all stable isotopes.

**Response:** We are sorry for the unclear expression; we have changed "enriched" to "heavier" in the text.

5. The lines marked "dissolution" in Fig. 6 are not defined. In the text, it mentions mineral dissolution but does not say which mineral.

**Response:** Thank you for the reminder. Accordingly, we have added explanations in "P19 RL9 to 10" and "P20 RL6 to 7".